# Complete Graphical Criterion for Sequential Covariate Adjustment in Causal Inference

**Yonghan Jung**[1*]   **Min Woo Park**[2]   **Sanghack Lee**[2*]
[1]Purdue University    [2]Seoul National University
jung222@purdue.edu   {alsdn0110,sanghack}@snu.ac.kr

## Abstract

Covariate adjustment, also known as back-door adjustment, is a fundamental tool in causal inference. Although a sound and complete graphical identification criterion, known as adjustment criterion (Shpitser et al., 2010), exists for static contexts, sequential contexts present challenges. Current practices, such as the sequential back-door adjustment (Pearl and Robins, 1995) or multi-outcome sequential back-door adjustment (Jung et al., 2020), are sound but incomplete; i.e., there are graphical scenarios where the causal effect is expressible via covariate adjustment, yet these criteria do not cover. In this paper, we exemplify this incompleteness and then present the *sequential adjustment criterion*, a sound and complete criterion for sequential covariate adjustment. We provide a constructive sequential adjustment criterion that identifies a set that satisfies the sequential adjustment criterion if and only if the causal effect can be expressed as a sequential covariate adjustment. Finally, we present an algorithm for identifying a *minimal* sequential covariate adjustment set, which optimizes efficiency by ensuring that no unnecessary vertices are included.

## 1   Introduction

Covariate adjustment (also known as the back-door adjustment (Pearl, 1995)) —$\sum_{\mathbf{z}} P(y \mid x, \mathbf{z}) P(\mathbf{z})$ where $\mathbf{Z}$ is a set of covariates, $X$ is a treatment, and $Y$ is an outcome—is one of the most prevalent method for estimating causal effects from observational data. Sufficient conditions have been established to determine whether causal effects can be expressed through covariate adjustment. For example, back-door criterion (BD) (Pearl, 1995) is a sound graphical condition to determine if the causal effect is expressible as a covariate adjustment. Shpitser et al. (2010) extended the BD and developed a sound and complete graphical criterion, called adjustment criterion (AC). van der Zander et al. (2014) and Maathuis and Colombo (2015) extended the adjustment criterion to general graph classes including directed acyclic graphs, maximal ancestral graphs, or partial ancestral graphs. The criterion from Maathuis and Colombo (2015) is sound but incomplete, which Perkovic et al. (2018) addressed by developing a sound and complete adjustment criterion.

Beyond such static adjustment criterion, *sequential* covariate adjustment (also known as the *g-formula* (Robins, 1986)) has been studied. For example, *sequential back-door (SBD) criterion* (Pearl and Robins, 1995) extended the back-door criterion to the sequential setting. Since the SBD criterion only considered a single outcome, Jung et al. (2020) extended it to accommodate multiple outcome variables called *multi-outcome sequential back-door (mSBD)* criterion. However, as detailed in Sec. 3, this criterion is not complete; i.e., there exists an example where the mSBD criterion is not satisfied, but the causal effect is identified as a sequential covariate adjustment. While it is known that the *sequential ignorability condition* (Richardson and Robins, 2013) is sound and complete for

---

*Corresponding authors

Table 1: Comparison of graphical criteria for (sequential) covariate adjustments

| Criterion | Static | Sequential | Multi-outcome | Completeness |
|---|---|---|---|---|
| BD (Pearl, 1995, 2000) | ✓ | ✗ | N/A | ✗ |
| AC (Shpitser et al., 2010) | ✓ | ✗ | N/A | ✓ |
| SBD (Pearl and Robins, 1995) | ✓ | ✓ | ✗ | ✗ |
| mSBD (Jung et al., 2020) | ✓ | ✓ | ✓ | ✗ |
| SAC (Ours) | ✓ | ✓ | ✓ | ✓ |

the g-formula estimand, it does not provide a graphical criterion that can be tested based on a graph compatible with the distribution.

To address this challenge, we devise a sound and complete graphical criterion for the sequential covariate adjustment. Table 1 visualizes our contribution against previous graphical criteria for covariate adjustments.[2] Specifically, our contributions are as follows:

1. We demonstrate the incompleteness of mSBD criterion by exemplifying a graph such that the causal effect is given as sequential covariate adjustment where the mSBD criterion is not satisfied. Moreover, we discuss the impracticality of other existing criteria.

2. We devise sequential adjustment criterion (SAC), a sound and complete graphical criterion for sequential covariate adjustment.

3. We develop a constructive sequential adjustment criterion that yields a set that satisfies the SAC if and only if the causal effect can be expressed as a sequential covariate adjustment. Furthermore, we present an algorithm to identify a minimal sequential covariate adjustment set.

The proofs for all results are provided in Sec. A in the supplementary document.

## 2   Preliminaries

We introduce the necessary notation and background. Key notations are summarized in Table 2.

**Structural Causal Models.**   We use Structural Causal Models (SCMs) (Pearl, 2000). An SCM $\mathcal{M}$ is a quadruple $\mathcal{M} \coloneqq \langle \mathbf{U}, \mathbf{V}, P(\mathbf{U}), F \rangle$. $\mathbf{U}$ is a set of exogenous (latent) variables following a joint distribution $P(\mathbf{U})$. $\mathbf{V}$ is a set of endogenous (observable) variables whose values are determined by functions $F = \{f_{V_i}\}_{V_i \in \mathbf{V}}$ such that $V_i \leftarrow f_{V_i}(\mathbf{pa}_i, \mathbf{u}_i)$ where $\mathbf{PA}_i \subseteq \mathbf{V} \setminus \{V_i\}$ and $\mathbf{U}_i \subseteq \mathbf{U}$. Each SCM $\mathcal{M}$ induces a distribution $P(\mathbf{V})$ and a causal graph $\mathcal{G} = \mathcal{G}(\mathcal{M})$ over $\mathbf{V}$ in which there exists a directed edge from every variable in $\mathbf{PA}_i$ to $V_i$ and dashed-bidirected arrows encode correlated latent variables. Performing an intervention fixing $\mathbf{X} = \mathbf{x}$ is represented through the do-operator, $\mathrm{do}(\mathbf{X} = \mathbf{x})$, which encodes the operation of replacing the original equations of $X$ (i.e., $f_X(\mathbf{pa}_x, \mathbf{u}_x)$) by the constant $x$ for all $X \in \mathbf{X}$ and induces an interventional distribution $P(\mathbf{V} \mid \mathrm{do}(\mathbf{x}))$.

**Graphs.**   We consider a graph $\mathcal{G}$ having vertices $\mathbf{V}$ and edges $\mathbf{E}$ composed of directed ($V_i \rightarrow V_j$) and bidirected edges ($V_i \leftrightarrow V_j$). An ordered sequence of distinct nodes in $\mathcal{G}$ is called a *path* between $V_i$ and $V_j$ in $\mathcal{G}$ if (1) the start node is $V_i$ and the end node is $V_j$, and (2) there is an edge (directed or bidirected) between any two subsequent variables in the sequence. A variable $V_i$ is called a *collider* on the path if the path contains two edges having arrow heads toward $V_i$; i.e., $* \rightarrow V_i \leftarrow *$ where $* \rightarrow \in \{\rightarrow, \leftrightarrow\}$. A path is *directed* (or causal), if it contains only directed edges, all pointing in the same direction. We say that two sets of vertices $\mathbf{A}, \mathbf{B} \subseteq \mathbf{V}$ are *d-connected* w.r.t. $\mathbf{Z} \subseteq \mathbf{V} \setminus (\mathbf{A} \cup \mathbf{B})$ in $\mathcal{G}$ if there exists a path between $V_i \in \mathbf{A}$ and $V_j \in \mathbf{B}$ where every non-colliders on the path is not contained in $\mathbf{Z}$ and all colliders are in $\mathrm{An}_{\mathcal{G}}(\mathbf{Z})$. We say that $\mathbf{Z}$ *d-separates* $(\mathbf{A}, \mathbf{B})$ in $\mathcal{G}$ if they are not d-connected w.r.t. $\mathbf{Z}$ in $\mathcal{G}$. We denote $\mathcal{G}_{\overline{\mathbf{A}}\underline{\mathbf{B}}}$ by $\mathcal{G}$ with removing those edges toward $\mathbf{A}$ and from $\mathbf{B}$. For disjoint sets of vertices $\mathbf{A}_1, \cdots, \mathbf{A}_m \subseteq \mathbf{A}$, we use $(\mathbf{A}_1, \cdots, \mathbf{A}_m)$ to represent an ordered set, where the order needs to be highlighted. For the unordered set, we use braces as $\{A_1, \cdots, A_m\}$.

---

[2]The multi-outcome column indicates whether the outcome for each criterion is a sequence of vectors or a single vector. Therefore, BD and AC, which deal with static context, are not applicable.

Table 2: Key notations

| Notation | Definition |
|---|---|
| $\mathbf{X}, X, \mathbf{x}, x$ | A random vector, variable, and their realized values. |
| $\sum_{\mathbf{x}} f(\mathbf{x})$ | A marginalization of $f(\mathbf{x})$ over values in the domain of $\mathbf{X}$. |
| $\mathbf{X}^{(i)}$ | $(\mathbf{X}_0, \cdots, \mathbf{X}_i)$ for the ordered set $\mathbf{X} = (\mathbf{X}_0, \cdots, \mathbf{X}_n)$. |
| $\mathbf{X}^{\geq i}$ | $(\mathbf{X}_i, \cdots, \mathbf{X}_n)$ for the ordered set $\mathbf{X}$. |
| $\mathrm{De}_{\mathcal{G}}(\mathbf{C}), \mathrm{An}_{\mathcal{G}}(\mathbf{C})$ | Descendants and ancestors of $\mathbf{C}$ in $\mathcal{G}$. |
| $\mathrm{pcp}_{\mathcal{G}}(\mathbf{X}, \mathbf{Y})$ | Proper causal path set; i.e., $(\mathrm{De}_{\mathcal{G}_{\overline{\mathbf{X}}}}(\mathbf{X}) \setminus \mathbf{X}) \cap \mathrm{An}_{\mathcal{G}_{\underline{\mathbf{X}}}}(\mathbf{Y})$. |
| $\mathrm{dpcp}_{\mathcal{G}}(\mathbf{X}, \mathbf{Y})$ | $\mathrm{De}_{\mathcal{G}}(\mathrm{pcp}_{\mathcal{G}}(\mathbf{X}, \mathbf{Y}))$. |

**Adjustment Criterion.** We revisit the problem of characterizing a criterion such that the causal effect is given as an adjustment. Formally, the *adjustment set* relative to $(\mathbf{X}, \mathbf{Y})$ in $\mathcal{G}$ is defined as a set $\mathbf{Z}$ such that, for any $P$ compatible with $\mathcal{G}$, the following is satisfied (Shpitser et al., 2010):

$$P(\mathbf{y} \mid \mathrm{do}(\mathbf{x})) = \sum_{\mathbf{z}} P(\mathbf{y} \mid \mathbf{x}, \mathbf{z}) P(\mathbf{z}). \tag{1}$$

The *adjustment criterion* (Shpitser et al., 2010) provides a sound and complete criterion for the causal effect $P(\mathbf{y} \mid \mathrm{do}(\mathbf{x}))$ to be expressed as a covariate adjustment in Eq. (1). To present the adjustment criterion, we define a *proper causal path* between $\mathbf{X}$ and $\mathbf{Y}$ as a directed path from any of $X \in \mathbf{X}$ to any of $Y \in \mathbf{Y}$ on which the path does not contain a node in $\mathbf{X} \setminus \{X\}$. A proper causal path set $\mathrm{pcp}_{\mathcal{G}}(\mathbf{X}, \mathbf{Y})$ is a collection of nodes excluding $\mathbf{X}$ on all proper causal paths from $X \in \mathbf{X}$ to $Y \in \mathbf{Y}$, defined in Table 2. $\mathrm{dpcp}_{\mathcal{G}}(\mathbf{X}, \mathbf{Y})$ is a descendant of the proper causal path set defined in Table 2. The adjustment criterion utilizes this subgraph called the *proper back-door graph*:

**Definition 1** (Proper Back-Door Graph (van der Zander et al., 2014)). *Let $(\mathbf{X}, \mathbf{Y})$ denote a disjoint pair in $\mathbf{V}$. The proper back-door graph $\mathcal{G}_{\mathrm{pbd}}^{\mathbf{X}, \mathbf{Y}}$ is a graph obtained from $\mathcal{G}$ by removing an edge $X \to D$ for each $X \in \mathbf{X}$ and $D \in \mathrm{pcp}_{\mathcal{G}}(\mathbf{X}, \mathbf{Y})$.*

Then, the adjustment criterion and its characterization are given as follows.

**Definition 2** (Adjustment Criterion (Shpitser et al., 2010; van der Zander et al., 2014)). *Let $(\mathbf{X}, \mathbf{Y})$ denote a disjoint pair in $\mathbf{V}$. A set of variables $\mathbf{Z} \subseteq \mathbf{V} \setminus (\mathbf{X} \cup \mathbf{Y})$ is said to satisfy the adjustment criterion relative to $(\mathbf{X}, \mathbf{Y})$ in $\mathcal{G}$ if (1) $(\mathbf{Y} \perp\!\!\!\perp \mathbf{X} \mid \mathbf{Z})_{\mathcal{G}_{\mathrm{pbd}}^{\mathbf{X}, \mathbf{Y}}}$ and (2) $\mathbf{Z} \cap \mathrm{dpcp}_{\mathcal{G}}(\mathbf{X}, \mathbf{Y}) = \emptyset$.*

**Proposition 1** (Shpitser et al. (2010); van der Zander et al. (2014)). *$\mathbf{Z}$ satisfies the adjustment criterion relative to $(\mathbf{X}, \mathbf{Y})$ in $\mathcal{G}$ if and only if $\mathbf{Z}$ is an adjustment set relative to $(\mathbf{X}, \mathbf{Y})$ in $\mathcal{G}$.*

For concreteness, in Fig. 3, $\mathbf{Z} := \{Z_a, Z_b\}$ satisfies the adjustment criterion relative to $\mathbf{X} := \{X_1, X_2\}$ and $\mathbf{Y} := \{Y_1, Y_2\}$. See (van der Zander et al., 2014, Sec. 4.2) for details. In the later sections, we will extend this result to the sequential context.

## 3 Limitations of Existing Criterion for Sequential Covariate Adjustment

In this section, we demonstrate the limitations of existing criteria for sequential covariate adjustment, which is defined as follows:

**Definition 3** (Sequential Covariate Adjustment). *Let $(\mathbf{X}, \mathbf{Y})$ denote a pair of ordered sets such that $\mathbf{X} = (\mathbf{X}_1, \ldots, \mathbf{X}_m)$ and $\mathbf{Y} = (\mathbf{Y}_0, \ldots, \mathbf{Y}_m)$. Let $\mathbf{Z} \subseteq \mathbf{V} \setminus (\mathbf{X} \cup \mathbf{Y})$ denote vertices ordered as $\mathbf{Z} = (\mathbf{Z}_1, \ldots, \mathbf{Z}_m)$. Define $\mathbf{H}_i := \mathbf{X}^{(i)} \cup \mathbf{Y}^{(i)} \cup \mathbf{Z}^{(i)}$. The set $\mathbf{Z}$ is said to be a sequential adjustment set relative to $(\mathbf{X}, \mathbf{Y})$ in $\mathcal{G}$ if for any $P$ compatible with $\mathcal{G}$, the following is satisfied[3]:*

$$P(\mathbf{y} \mid \mathrm{do}(\mathbf{x})) = \sum_{\mathbf{z}} \prod_{j=0}^{m} P(\mathbf{z}_{j+1}, \mathbf{y}_j \mid \mathbf{h}_{j-1}, \mathbf{x}_j, \mathbf{z}_j). \tag{2}$$

*The right-hand side of Eq. (2) is called the sequential covariate adjustment of $\mathbf{Z}$ w.r.t. $(\mathbf{X}, \mathbf{Y})$ in $\mathcal{G}$.*

---

[3]If an ordered set associates with an index out of range (e.g., $\mathbf{X}_0, \mathbf{X}_{m+1}$), it is interpreted as the empty set. An interval $[a, b]$ (e.g., those associated with $\sum$ or $\bigcup$) with $b < a$ is also an empty set.

In this definition, sequential covariate adjustment relies on specific partitions $\mathbf{X} = (\mathbf{X}_1, \cdots, \mathbf{X}_m)$ and $\mathbf{Y} = (\mathbf{Y}_0, \cdots, \mathbf{Y}_m)$. To standardize these partitions, we employ a fixed structure based on a topological order of $(\mathbf{X}, \mathbf{Y})$ in $\mathcal{G}$ so as to ensure that the partition aligns with the causal structure.

**Definition 4** (Partitioning Operator). *Let $\mathbf{X}$ and $\mathbf{Y}$ denote disjoint vertices where $\mathbf{X} = (X_1, \cdots, X_m)$ is topologically ordered along with $\mathcal{G}$. Then, the operator $\mathrm{PT}_{\mathcal{G}}^{\mathbf{X}}$ partitions $\mathbf{Y}$ relative to $\mathbf{X}$ and $\mathcal{G}$; i.e., $\mathrm{PT}_{\mathcal{G}}^{\mathbf{X}}(\mathbf{Y}) = (\mathbf{Y}_0, \cdots, \mathbf{Y}_m)$, where*

$$\mathbf{Y}_i \coloneqq \begin{cases} \mathbf{Y} \setminus \mathrm{De}_{\mathcal{G}}(\mathbf{X}) & \text{for } i = 0, \\ \left( (\mathbf{Y} \setminus \mathbf{Y}^{(i-1)}) \cap \mathrm{De}_{\mathcal{G}}(X_i) \right) \setminus \mathrm{De}_{\mathcal{G}}(\mathbf{X}^{\geq i+1}) & \text{for } i = 1, \cdots, m. \end{cases} \tag{3}$$

Hereafter, we will denote $(\mathbf{X}, \mathbf{Y})$ as a pair of vertices where $\mathbf{X} = (X_1, \cdots, X_m)$ is topologically ordered with respect to $\mathcal{G}$ and $\mathbf{Y}$ as the ordered set $\mathrm{PT}_{\mathcal{G}}^{\mathbf{X}}(\mathbf{Y})$.

### 3.1 Limitation of mSBD Criterion

The multi-outcome sequential back-door (mSBD) criterion (Jung et al., 2020) has been developed to determine whether an ordered set $\mathbf{Z} = (\mathbf{Z}_1, \cdots, \mathbf{Z}_m)$ constitutes a sequential adjustment set:

**Definition 5** (Multi-outcome Sequential Back-Door Criterion (Jung et al., 2020)). *Let $(\mathbf{X}, \mathbf{Y})$ denote a pair of ordered sets in $\mathcal{G}$. Let $\mathbf{Z} \subseteq \mathbf{V} \setminus (\mathbf{X} \cup \mathbf{Y})$ be ordered as $\mathbf{Z} = (\mathbf{Z}_1, \ldots, \mathbf{Z}_m)$, and let $\mathbf{H}_i \coloneqq \mathbf{X}^{(i)} \cup \mathbf{Y}^{(i)} \cup \mathbf{Z}^{(i)}$. Then, $\mathbf{Z}$ is said to satisfy the multi-outcome sequential back-door (mSBD) criterion w.r.t. $(\mathbf{X}, \mathbf{Y})$ in $\mathcal{G}$ if, for $i = 1, \ldots, m$,*

$$\mathbf{Z}_i \text{ is non-descendant of } \mathbf{X}^{\geq i} \text{ in } \mathcal{G}; \text{ and} \tag{4}$$

$$\left( \mathbf{Y}^{\geq i} \perp\!\!\!\perp X_i \mid \mathbf{H}_{i-1}, \mathbf{Z}_i \right)_{\mathcal{G}_{\underline{X_i} \overline{\mathbf{X}^{\geq i+1}}}}. \tag{5}$$

**Proposition 2** (mSBD adjustment (Jung et al., 2020)). *If $\mathbf{Z} = (\mathbf{Z}_1, \ldots, \mathbf{Z}_m)$ satisfies the mSBD criterion w.r.t. $(\mathbf{X}, \mathbf{Y})$ in $\mathcal{G}$, the causal effect $P(\mathbf{y} \mid \mathrm{do}(\mathbf{x}))$ is identifiable and expressible as sequential covariate adjustment of $\mathbf{Z}$ w.r.t. $(\mathbf{X}, \mathbf{Y})$ in $\mathcal{G}$:*

$$P(\mathbf{y} \mid \mathrm{do}(\mathbf{x})) = \sum_{\mathbf{z}} \prod_{j=0}^{m} P(\mathbf{z}_{j+1}, \mathbf{y}_j \mid \mathbf{h}_{j-1}, x_j, \mathbf{z}_j). \tag{6}$$

For concreteness, consider the causal diagram $\mathcal{G}$ in Fig. 1 with $\mathbf{X} = (X_1, X_2)$ and $\mathbf{Y} = (\mathbf{Y}_1, \mathbf{Y}_2)$. Applying the partitioning operator $\mathrm{PT}_{\mathcal{G}}^{\mathbf{X}}$, we divide $\mathbf{Y}$ into two subsets: $\mathbf{Y}_1 = \{Y_1\}$ and $\mathbf{Y}_2 = \{Y_2\}$. Next, we define $\mathbf{Z} \coloneqq (\mathbf{Z}_1, \mathbf{Z}_2)$, where $\mathbf{Z}_1 \coloneqq \{Z_a, Z_b\}$ and $\mathbf{Z}_2 \coloneqq \{Z_c, Z_d\}$. This ordered set $\mathbf{Z}$ satisfies the mSBD criterion relative to $(\mathbf{X}, \mathbf{Y})$ in $\mathcal{G}$. The first condition of the mSBD criterion holds since $\mathbf{Z}_1$ does not include any descendants of $X_1$ in $\mathcal{G}$, and similarly, $\mathbf{Z}_2$ excludes any descendants of $X_2$. To check the second condition of the mSBD criterion, consider the graph $\mathcal{G}_{\underline{X_1}\overline{X_2}}$ derived from $\mathcal{G}$ by removing the edges $X_1 \to Y_1$ and $\{Z_b \to X_2, Z_c \to X_2, Y_1 \to X_2\}$. In this graph, $(\mathbf{Y}_1, \mathbf{Y}_2)$ are conditionally independent of $X_1$ given $\mathbf{Z}_1$

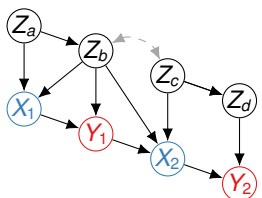

Figure 1: Graph exemplifying mSBD in Def. 5

and $X_2$, because any path from $X_1$ to $\{Y_1, Y_2\}$ is blocked (d-separated) by $\mathbf{Z}_1$. Finally, in the graph $\mathcal{G}_{\underline{X_2}}$, where the edge $X_2 \to Y_2$ is removed, any paths between $X_2$ and $Y_2$ are d-separated when conditioned on $\mathbf{Z}_1 \cup \mathbf{Z}_2$. This analysis confirms that $\mathbf{Z}$ satisfies the mSBD criterion, and therefore, the causal effect is given as follows:

$$P(\mathbf{y} \mid \mathrm{do}(\mathbf{x})) = \sum_{\mathbf{z}_1, \mathbf{z}_2} P(\mathbf{y}_2 \mid \mathbf{x}_1, \mathbf{x}_2, \mathbf{y}_1, \mathbf{z}_1, \mathbf{z}_2) P(\mathbf{z}_2, \mathbf{y}_1 \mid \mathbf{x}_1, \mathbf{z}_1) P(\mathbf{z}_1)$$

$$= \sum_{z_a, z_b, z_c, z_d} P(y_2 \mid x_1, x_2, y_1, z_a, z_b, z_c, z_d) P(z_c, z_d, y_1 \mid x_1, z_a, z_b) P(z_a, z_b). \tag{7}$$

Despite its soundness (Prop. 2), the mSBD criterion is not complete. In other words, there are causal graphs such that $P(\mathbf{y} \mid \mathrm{do}(\mathbf{x}))$ is expressible through sequential covariate adjustment, even though $\mathbf{Z} \coloneqq (\mathbf{Z}_1, \ldots, \mathbf{Z}_m)$ does not satisfy the mSBD criterion relative to $(\mathbf{X}, \mathbf{Y})$ in $\mathcal{G}$.

**Proposition 3** (Incompleteness of mSBD criterion). *There exists a causal graph $\mathcal{G}$ where the causal effect $P(\mathbf{y} \mid \mathrm{do}(\mathbf{x}))$ can be represented as a sequential covariate adjustment with $\mathbf{Z}$ such that $\mathbf{Z}$ does not satisfy the mSBD criterion relative to $(\mathbf{X}, \mathbf{Y})$ in $\mathcal{G}$.*

To illustrate the incompleteness of the mSBD criterion, consider the graph $\mathcal{G}$ in Fig. 2a with $\mathbf{X} = (X_1, X_2)$ and $\mathbf{Y} = (\mathbf{Y}_1, \mathbf{Y}_2)$ (i.e., $\mathbf{Y}_0 := \emptyset$) with $\mathbf{Y}_1 = \{Y_1\}$ and $\mathbf{Y}_2 = \{Y_2\}$. Set $\mathbf{Z} := (\mathbf{Z}_1, \mathbf{Z}_2)$, where $\mathbf{Z}_1 := \{Z_a, Z_b\}$ and $\mathbf{Z}_2 := \{Z_c, Z_d\}$. Then, while $\mathbf{Z}_1$ satisfies both Eqs. (4, 5) of the mSBD criterion, $\mathbf{Z}_2$ does not satisfy the mSBD condition in Eq. (4) since $\mathbf{Z}_2$ includes the descendant of $X_2$. Nevertheless, the causal effect $P(\mathbf{y} \mid \mathrm{do}(\mathbf{x}))$ is still expressible through sequential covariate adjustment as in Eq. (7). To demonstrate, we begin with the observation that $\mathbf{Z}_1$ satisfies the mSBD criterion in Eqs. (4, 5), leading to $P(y_1, y_2 \mid \mathrm{do}(x_1, x_2)) = \sum_{z_a, z_b} P(y_2 \mid \mathrm{do}(x_2), \mathbf{h}_1) P(y_1 \mid x_1, z_a, z_b) P(z_a, z_b)$, with $\mathbf{H}_1 := \mathbf{X}_1 \cup \mathbf{Y}_1 \cup \mathbf{Z}_1 = \{X_1, Y_1, Z_a, Z_b\}$. Next,

$$P(y_2 \mid \mathrm{do}(x_2), \mathbf{h}_1) = \sum_{z_c} P(y_2 \mid \mathrm{do}(x_2), \mathbf{h}_1) P(z_c \mid \mathbf{h}_1)$$

$$= \sum_{z_c} P(y_2 \mid \mathrm{do}(x_2), \mathbf{h}_1, z_c) P(z_c \mid \mathbf{h}_1),$$

where the last equality holds by Rule 1 of do-calculus (Pearl, 1995): $(Y_2 \perp\!\!\!\perp Z_c \mid X_2, \mathbf{H}_1)_{\mathcal{G}_{\overline{X_2}}}$. Further,

$$P(y_2 \mid \mathrm{do}(x_2), \mathbf{h}_1, z_c) = \sum_{z_d} P(y_2 \mid \mathrm{do}(x_2), \mathbf{h}_1, z_c, z_d) P(z_d \mid \mathrm{do}(x_2), \mathbf{h}_1, z_c)$$

$$= \sum_{z_d} P(y_2 \mid x_2, \mathbf{h}_1, z_c, z_d) P(z_d \mid \mathbf{h}_1, z_c),$$

where the last equality holds by Rule 2 and 3 of do-calculus, i.e., $(Y_2 \perp\!\!\!\perp X_2 \mid \mathbf{H}_1, Z_c, Z_d)_{\mathcal{G}_{\underline{X_2}}}$ and $(Z_d \perp\!\!\!\perp X_2 \mid \mathbf{H}_1, Z_c)_{\mathcal{G}_{\overline{X_2}}}$, respectively. Combining these derivations, $P(y_1, y_2 \mid \mathrm{do}(x_1, x_2))$ is expressible as the same as in Eq. (7). This exemplifies the incompleteness of mSBD criterion.

### 3.2 Limitation of Other Existing Criteria

Besides mSBD, there are other methods attempting to establish conditions for determining if the causal effect can be represented as sequential covariate adjustment. For instance, the *no-unmeasured-confounding assumption* (Rosenbaum and Rubin, 1983) is a sufficient verification tool. However, as illustrated in Fig. 2a, the causal effect is expressible through sequential covariate adjustment even though an unmeasured confounder between the treatment $X_2$ and the outcome $Y_1$ presents.

On the other hand, the *sequential ignorability* criterion (Robins, 1986, 2000) provides a condition for $\mathbf{Z}$ to be a sequential adjustment set:

$$\mathbf{Y}^{\geq i}(\mathbf{x}) \perp\!\!\!\perp X_i \mid \mathbf{Z}_i, \mathbf{H}_{i-1}, \tag{8}$$

where $\mathbf{Y}^{\geq i}(\mathbf{x})$ is a counterfactual variable induced from $\mathbf{Y}^{\geq i}$ by fixing $\mathbf{X} = \mathbf{x}$ in the SCM. Graphical methods such as *single world intervention graphs* (SWIGs) (Richardson and Robins, 2013) and *twin-networks* (Pearl, 2000; Shpitser and Pearl, 2008) help in analyzing such counterfactual variables. However, each has its limitations in efficiently verifying sequential adjustment sets. SWIGs, for example, face challenges in capturing $\mathbf{Z}_1, \mathbf{Z}_2$ as a sequential adjustment set in Fig 2a, because they fail to accommodate both $Y_2(x_2)$ and $\mathbf{Z}_2$ in a SWIG. This limitation arises since the counterfactual variable $\mathbf{Z}_2(x_2)$, instead of the observable $\mathbf{Z}_2$, will appear on the SWIG. In contrast, twin-networks are capable of exploring the relationship between $Y_2(x_2)$ and $\mathbf{Z}_2$. Yet, their practicality decreases with larger cardinality ($m$), as the size of twin-networks expands exponentially, leading to substantial computational demands. These observations motivate the necessity of computationally efficient and effective criterion to determine if a causal effect aligns with a sequential covariate adjustment.

## 4 Sequential Adjustment Criterion

In this section, we devise a *sequential adjustment criterion* (SAC), which provides a sound and complete method for characterizing the causal effect $P(\mathbf{y} \mid \mathrm{do}(\mathbf{x}))$ as a sequential covariate adjustment of $\mathbf{Z} := (\mathbf{Z}_1, \cdots, \mathbf{Z}_m)$ with respect to $(\mathbf{X}, \mathbf{Y})$. To begin with, we define a special subgraph called *proper sequential back-door graph*:

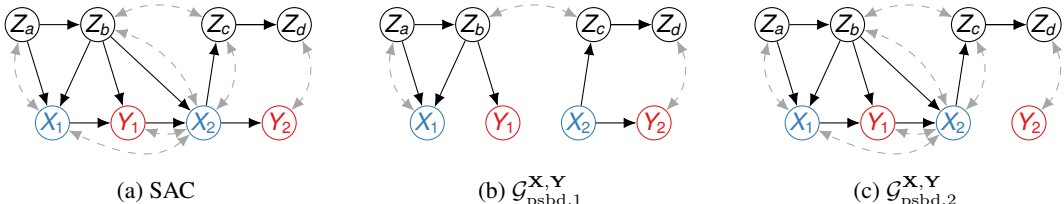

(a) SAC      (b) $\mathcal{G}^{\mathbf{X},\mathbf{Y}}_{\mathrm{psbd},1}$      (c) $\mathcal{G}^{\mathbf{X},\mathbf{Y}}_{\mathrm{psbd},2}$

Figure 2: (a) Graph exemplifying SAC in Def. 7 and (b,c) its proper sequential back-door graphs

**Definition 6** (Proper Sequential Back-Door Graph). *For a disjoint pair* $(\mathbf{X}, \mathbf{Y})$ *and the index* $i$, *the proper sequential back-door graph* $\mathcal{G}^{\mathbf{X},\mathbf{Y}}_{\mathrm{psbd},i}$ *is a proper back-door graph induced from* $\mathcal{G}_{\overline{\mathbf{X}^{\geq i+1}}}$ *relative to* $(X_i, \mathbf{Y}^{\geq i})$; *i.e.,*

$$\mathcal{G}^{\mathbf{X},\mathbf{Y}}_{\mathrm{psbd},i} := (\mathcal{G}_{\overline{\mathbf{X}^{\geq i+1}}})^{X_i, \mathbf{Y}^{\geq i}}_{\mathrm{pbd}}. \tag{9}$$

As constructing the proper back-door graph from $\mathcal{G}$ takes linear time $O(|\mathbf{V}| + |\mathbf{E}|)$ (van der Zander et al., 2014), the sequential back-door graph can also be constructed in linear time $O(|\mathbf{V}| + |\mathbf{E}|)$. Equipped with $\mathcal{G}^{\mathbf{X},\mathbf{Y}}_{\mathrm{psbd},i}$, the *sequential adjustment criterion* is presented as follows:

**Definition 7** (Sequential Adjustment Criterion (SAC)). *Let* $(\mathbf{X}, \mathbf{Y})$ *denote a disjoint pair. Let* $\mathbf{Z} := (\mathbf{Z}_1, \cdots, \mathbf{Z}_m)$ *denote a topologically ordered set of vertices disjoint to* $(\mathbf{X}, \mathbf{Y})$ *where each* $\mathbf{Z}_i$ *is non-descendant of* $\mathbf{X}^{\geq i+1}$. *Then,* $\mathbf{Z}$ *is said to satisfy sequential adjustment criterion (SAC) w.r.t.* $(\mathbf{X}, \mathbf{Y})$ *in* $\mathcal{G}$ *if the following conditions are satisfied for* $i = 1, \cdots, m$:

$$(\mathbf{Y}^{\geq i} \perp\!\!\!\perp X_i \mid \mathbf{Z}_i, \mathbf{H}_{i-1})_{\mathcal{G}^{\mathbf{X},\mathbf{Y}}_{\mathrm{psbd},i}}; \text{ and} \tag{10}$$

$$\mathbf{Z}_i \cap \mathrm{dpcp}_{\mathcal{G}}(X_i, \mathbf{Y}^{\geq i}) = \emptyset. \tag{11}$$

The sequential adjustment criterion characterizes the sequential covariate adjustment:

**Theorem 1** (Soundness and Completeness). *Let* $(\mathbf{X}, \mathbf{Y})$ *denote a disjoint pair, and let* $\mathbf{Z} := (\mathbf{Z}_1, \cdots, \mathbf{Z}_m)$ *denote an ordered set of vertices disjoint to* $(\mathbf{X}, \mathbf{Y})$. *Then, the following are equivalent:*

1. $\mathbf{Z}$ *satisfies SAC relative to* $(\mathbf{X}, \mathbf{Y})$ *in* $\mathcal{G}$.

2. $\mathbf{Z}$ *is a sequential adjustment set relative to* $(\mathbf{X}, \mathbf{Y})$ *in* $\mathcal{G}$; *i.e.,*

$$P(\mathbf{y} \mid \mathrm{do}(\mathbf{x})) = \sum_{\mathbf{z}} \prod_{j=0}^{m} P(\mathbf{z}_{j+1}, \mathbf{y}_j \mid \mathbf{h}_{j-1}, x_j, \mathbf{z}_j). \tag{12}$$

For instance, consider Fig. 2a, where we witnessed that the causal effect $P(y_1, y_2 \mid \mathrm{do}(x_1, x_2))$ is given as a sequential covariate adjustment of $\mathbf{Z} = (\mathbf{Z}_1 = \{Z_a, Z_b\}, \mathbf{Z}_2 = \{Z_c, Z_d\})$. To confirm that this $\mathbf{Z}$ satisfies SAC relative to $(\mathbf{X}, \mathbf{Y})$ for $\mathbf{X} = (X_1, X_2)$ and $\mathbf{Y} = (Y_1, Y_2)$ with $\mathbf{Y}_1 = \{Y_1\}$ and $\mathbf{Y}_2 = \{Y_2\}$, we examine $\mathcal{G}^{\mathbf{X},\mathbf{Y}}_{\mathrm{psbd},1}$ (Fig. 2b) and $\mathcal{G}^{\mathbf{X},\mathbf{Y}}_{\mathrm{psbd},2}$ (Fig. 2c). In $\mathcal{G}^{\mathbf{X},\mathbf{Y}}_{\mathrm{psbd},1}$, we observe that $\mathbf{Z}_1 = \{Z_a, Z_b\}$ d-separates $X_1$ and $\{Y_1, Y_2\}$. Moreover, $\mathbf{Z}_1$ does not contain any descendant of proper causal paths from $X_1$ to $\{Y_1, Y_2\}$ within this subgraph. Similarly, in $\mathcal{G}^{\mathbf{X},\mathbf{Y}}_{\mathrm{psbd},2}$, the path between $X_2$ and $Y_2$ is d-separated conditioned on $\mathbf{Z}_2 = \{Z_c, Z_d\}$ and $\mathbf{H}_1 = \{Z_a, Z_b, X_1, Y_1\}$. Additionally, $\mathbf{Z}_2$ is not on any descendant of proper causal paths between $X_2$ and $Y_2$. Consequently, $\mathbf{Z}$ fulfills SAC relative to $(\mathbf{X}, \mathbf{Y})$ in the graph $\mathcal{G}$.

The soundness and completeness of SAC provide extensive coverage, incorporating existing covariate adjustment criterion such as BD, AC, SBD, and mSBD in Table 1. Clearly, SAC covers the mSBD criterion due to its completeness. In other words, the mSBD criterion is sufficient for the sequential adjustment criterion.

**Corollary 1** (mSBD $\implies$ SAC). *If* $\mathbf{Z}$ *satisfies the mSBD criterion relative to* $(\mathbf{X}, \mathbf{Y})$ *in* $\mathcal{G}$, *then* $\mathbf{Z}$ *satisfies the SAC relative to* $(\mathbf{X}, \mathbf{Y})$ *in* $\mathcal{G}$.

To demonstrate, consider $\mathcal{G}$ in Fig. 1, where $\mathbf{Z} := (\mathbf{Z}_1, \mathbf{Z}_2)$ with $\mathbf{Z}_1 := \{Z_a, Z_b\}$ and $\mathbf{Z}_2 := \{Z_c, Z_d\}$. This $\mathbf{Z}$ satisfies the mSBD criterion relative to $(\mathbf{X}, \mathbf{Y})$ for $\mathbf{X} = (X_1, X_2)$ and $\mathbf{Y} = (Y_1, Y_2)$. To

ensure that $\mathbf{Z}$ also satisfies the SAC relative to $(\mathbf{X}, \mathbf{Y})$ in $\mathcal{G}$, we verify that $(\mathbf{Z}_1, \mathbf{Z}_2)$ meet the conditions in Eqs. (10, 11), respectively. Since $\mathbf{Z}_1$ and $\mathbf{Z}_2$ are both non-descendants of $X_1$ and $X_2$, respectively, we only need to verify that they meet the conditions of Eq. (11) for $(X_1, (Y_1, Y_2))$ and $(X_2, Y_2)$ respectively. First, consider $\mathcal{G}_{\text{psbd},1}^{\mathbf{X},\mathbf{Y}}$, which removes edges $\{X_1 \to Y_1, Z_b \to X_2, Z_c \to X_2\}$ from $\mathcal{G}$. In this graph, $X_1$ is d-separated from $(Y_1, Y_2)$ given $\mathbf{Z}_1$. Next, consider $\mathcal{G}_{\text{psbd},2}^{\mathbf{X},\mathbf{Y}}$, which removes the edge $\{X_2 \to Y_2\}$ from $\mathcal{G}$. In this graph, $X_2$ is d-separated from $Y_2$ given $(\mathbf{Z}_1, X_1, Y_1)$ and $\mathbf{Z}_2$. Therefore, $\mathbf{Z}$ also satisfies the SAC relative to $(\mathbf{X}, \mathbf{Y})$ in $\mathcal{G}$.

The broader coverage of SAC compared to the mSBD criterion is evident when compared to AC. First, the mSBD criterion is not extensive enough to cover the AC criterion. To witness, consider Fig. 3. Here, $\mathbf{Z} := \{Z_a, Z_b\}$ satisfies AC relative to $(\mathbf{X}, \mathbf{Y})$ for $\mathbf{X} = (X, Y)$ and $\mathbf{Y} = (Y_1, Y_2)$. However, $\mathbf{Z}$ does not satisfy the mSBD criterion since it is a descendant set of $\mathbf{X}_1 := \{X_1\}$. In contrast, AC is a sufficient criterion for SAC.

**Theorem 2** (AC $\implies$ SAC). *If $\mathbf{Z}$ satisfies the AC relative to $(\mathbf{X}, \mathbf{Y})$ in $\mathcal{G}$, then there exists a partition of $\mathbf{Z}$ and a topological order of $\mathbf{X}$ such that $\mathbf{Z}$ satisfies the SAC relative to $(\mathbf{X}, \mathbf{Y})$ in $\mathcal{G}$.*

For concreteness, consider $\mathcal{G}$ in Fig. 3 (van der Zander et al., 2014). In $\mathcal{G}$, $\mathbf{Z} := \{Z_a, Z_b\}$ satisfies the adjustment criterion relative to $(\mathbf{X}, \mathbf{Y})$ for $\mathbf{X} = (X_1, X_2)$ and $\mathbf{Y} = (Y_1, Y_2)$ in $\mathcal{G}$, and the causal effect is given as an adjustment as follows:

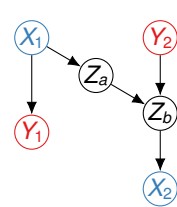

$$P(y_1, y_2 \mid \text{do}(x_1, x_2)) = \sum_{\mathbf{z}} P(\mathbf{y} \mid \mathbf{x}, \mathbf{z}) P(\mathbf{z}). \qquad (13)$$

Thm. 2 implies that $\mathbf{Z}$ meets the SAC relative to $(\mathbf{X}, \mathbf{Y})$ in graph $\mathcal{G}$. To demonstrate this, for $\mathbf{X} = (X_1, X_2)$, we apply the partitioning operator in Def. 4 and obtain $\mathbf{Y}_0 = \{Y_2\}$, $\mathbf{Y}_1 := \{Y_1\}$, and $\mathbf{Y}_2 = \emptyset$. We set

Figure 3: (van der Zander et al., 2014, Fig. 2)

$\mathbf{Z}_1 := \{Z_a, Z_b\}$. Since $\mathbf{Y}_2$ is empty, it is sufficient to demonstrate that $\mathbf{Z}_1$ meets the conditions of Eqs. (10) and (11). First, $\mathbf{Z}_1$ meets the condition of Eq. (11) as it contains no descendants on the causal paths from $X_1$ to $\mathbf{Y}_1$. Consider the modified graph $\mathcal{G}_{\text{psbd},1}^{\mathbf{X},\mathbf{Y}}$, obtained by removing the edges $\{X_1 \to Y_1, Z_b \to X_2\}$. In this reduced graph, $X_1$ and $\mathbf{Y}_1$ are d-separated given $\mathbf{Z}_1$ and $\mathbf{Y}_0$, confirming the conditions of SAC. Therefore, $\mathbf{Z} := (\mathbf{Z}_1, \mathbf{Z}_2)$ satisfies SAC relative to $(\mathbf{X}, \mathbf{Y})$ in $\mathcal{G}$, and the causal effect is described as

$$P(y_1, y_2 \mid \text{do}(x_1, x_2)) = \sum_{z_a, z_b} P(y_1 \mid x_1, z_a, z_b, y_2) P(z_a, z_b \mid y_2) P(y_2). \qquad (14)$$

Eqs. (13, 14) are equivalent since $P(y_1 \mid x_1, x_2, z_a, z_b, y_2) = P(y_1 \mid x_1, z_a, z_b, y_2)$ and $P(y_2 \mid x_1, x_2, z_a, z_b) = P(y_2 \mid z_a, z_b)$ by the conditional independence implied by $\mathcal{G}$.

### 4.1 Constructive Sequential Adjustment Criterion

Sequential adjustment criterion in Def. 7 relies on a predefined partition $\mathbf{Z} = (\mathbf{Z}_1, \cdots, \mathbf{Z}_m)$. A natural question is how to construct such ordered partition of the covariates. We propose a method to construct a sequential adjustment set, which characterizes sequential covariate adjustment in that the ordered set meets the SAC if and only if the causal effect can be expressed through a sequential covariate adjustment.

**Theorem 3** (Construction of Sequential Adjustment Set). *Let $(\mathbf{X}, \mathbf{Y})$ denote a disjoint pair in $\mathcal{G}$. Define $\mathbf{Z}^{\text{an}} := (\mathbf{Z}_1^{\text{an}}, \cdots, \mathbf{Z}_m^{\text{an}})$ and corresponding $(\mathbf{F}_1^{\text{an}}, \cdots, \mathbf{F}_m^{\text{an}})$ and $(\mathbf{H}_0^{\text{an}}, \cdots, \mathbf{H}_{m-1}^{\text{an}})$ alternatively as follows: For $i = 1, \cdots, m,$*

$$\mathbf{H}_{i-1}^{\text{an}} := \mathbf{X}^{(i-1)} \cup \mathbf{Y}^{(i-1)} \cup \bigcup_{j=1}^{i-1} \mathbf{Z}_j^{\text{an}}, \qquad (15)$$

$$\mathbf{F}_i^{\text{an}} := \mathbf{X} \cup \mathbf{Y} \cup \mathbf{H}_{i-1}^{\text{an}} \cup \text{dpcp}_{\mathcal{G}}(X_i, \mathbf{Y}^{\geq i}) \cup \text{De}_{\mathcal{G}}(\mathbf{X}^{\geq i+1}), \qquad (16)$$

$$\mathbf{Z}_i^{\text{an}} := \text{An}_{\mathcal{G}_{\text{psbd},i}^{\mathbf{X},\mathbf{Y}}}(\{X_i\} \cup \mathbf{Y}^{\geq i} \cup \mathbf{H}_{i-1}^{\text{an}}) \setminus \mathbf{F}_i^{\text{an}}. \qquad (17)$$

*Then, the following statements are equivalent:*

1. *There exists an ordered set $\mathbf{Z} := (\mathbf{Z}_1, \cdots, \mathbf{Z}_m)$ satisfying the sequential adjustment criterion w.r.t. $(\mathbf{X}, \mathbf{Y})$ in $\mathcal{G}$.*

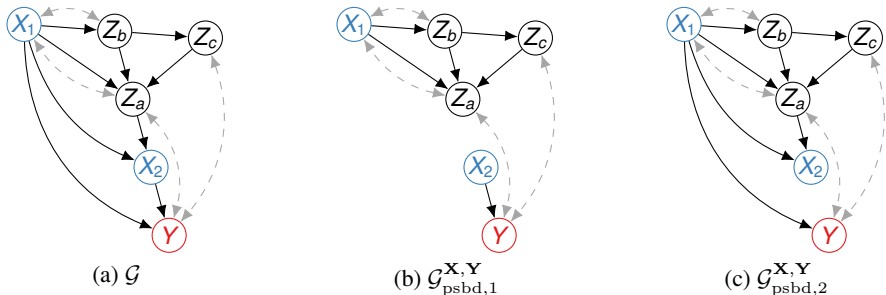

(a) $\mathcal{G}$  (b) $\mathcal{G}_{\mathrm{psbd},1}^{\mathbf{X},\mathbf{Y}}$  (c) $\mathcal{G}_{\mathrm{psbd},2}^{\mathbf{X},\mathbf{Y}}$

Figure 4: Example for `minSCA` and corresponding proper sequential back-door graphs

2. *The ordered set $\mathbf{Z}^{\mathrm{an}}$ satisfies SAC relative to $(\mathbf{X}, \mathbf{Y})$ in $\mathcal{G}$.*

The set $\mathbf{Z}^{\mathrm{an}}$ can be constructed in $O(m(|\mathbf{V}| + |\mathbf{E}|))$ since the procedures for constructing $\mathrm{dpcp}_{\mathcal{G}}(X_i, \mathbf{Y}^{\geq i})$ and $\mathcal{G}_{\mathrm{psbd},i}^{\mathbf{X},\mathbf{Y}}$ in Eq. (16), as well as finding ancestors and descendants in Eqs. (16, 17), each require $O(|\mathbf{V}| + |\mathbf{E}|)$ and are repeated $m$ times.

To concretely exemplify that the first statement implies the second one in Thm. 3, consider Fig. 2a. Set $\mathbf{Y}_0 := \emptyset$, $\mathbf{Y}_1 := \{Y_1\}$, and $\mathbf{Y}_2 := \{Y_2\}$ by the mechanism of the partitioning operator $\mathrm{PT}_{\mathcal{G}}^{\mathbf{X}}$. $\mathcal{G}_{\mathrm{psbd},1}^{\mathbf{X},\mathbf{Y}}$ and $\mathcal{G}_{\mathrm{psbd},2}^{\mathbf{X},\mathbf{Y}}$ are depicted in Figs. (2b, 2c). Then, $\mathbf{Z}_1^{\mathrm{an}} := \{Z_a, Z_b, Z_c, Z_d\}$, since these are ancestors of $\{X_1, Y_1, Y_2\}$ in $\mathcal{G}_{\mathrm{psbd},1}^{\mathbf{X},\mathbf{Y}}$. We can see that $\mathbf{Z}_1^{\mathrm{an}}$ satisfies Eqs. (10, 11) in the SAC because $X_1$ and $\{Y_1, Y_2\}$ are d-separated in $\mathcal{G}_{\mathrm{psbd},1}^{\mathbf{X},\mathbf{Y}}$ given $\mathbf{Z}_1^{\mathrm{an}}$, and they are not on the descendant of proper causal paths between $X_1$ and $\{Y_1, Y_2\}$. Next, $\mathbf{Z}_2^{\mathrm{an}} := \emptyset$. It satisfies Eqs. (10, 11) since $X_2$ and $Y_2$ are d-separated in $\mathcal{G}_{\mathrm{psbd},2}^{\mathbf{X},\mathbf{Y}}$ given $\{X_1, Y_1\} \cup \mathbf{Z}_1^{\mathrm{an}} \cup \mathbf{Z}_2^{\mathrm{an}}$. Combining with our previous observation that $\mathbf{Z}_1 := \{Z_a, Z_b\}$ and $\mathbf{Z}_2 := \{Z_c, Z_d\}$ satisfy Eqs. (10, 11), this example demonstrates that indeed $\mathbf{Z}^{\mathrm{an}} := (\mathbf{Z}_1^{\mathrm{an}}, \mathbf{Z}_2^{\mathrm{an}})$ satisfies the SAC relative to $(\mathbf{X}, \mathbf{Y})$ in $\mathcal{G}$.

As an example that the second statement implies the first one, consider Fig. 4a, where $\mathbf{Y}_0 := \emptyset$, $\mathbf{Y}_1 := \emptyset$ and $\mathbf{Y}_2 := \{Y\}$ by the mechanism of the partitioning operator $\mathrm{PT}_{\mathcal{G}}^{\mathbf{X}}$. Next, $\mathcal{G}_{\mathrm{psbd},1}^{\mathbf{X},\mathbf{Y}}$ and $\mathcal{G}_{\mathrm{psbd},2}^{\mathbf{X},\mathbf{Y}}$ are depicted in Figs. (4b, 4c). Then, $\mathbf{Z}_1^{\mathrm{an}} := \emptyset$, since none of $\{Z_a, Z_b, Z_c\}$ are the ancestors of $\{X_1, Y\}$. $\mathbf{Z}_1^{\mathrm{an}}$ satisfies Eqs. (10, 11) in the SAC, since the $X_1$ and $Y$ are d-separated in $\mathcal{G}_{\mathrm{psbd},1}^{\mathbf{X},\mathbf{Y}}$ given $\mathbf{Z}_1^{\mathrm{an}}$. Next, $\mathbf{Z}_2^{\mathrm{an}} := \{Z_a, Z_b, Z_c\}$ since they are ancestors of $X_2$ in $\mathcal{G}_{\mathrm{psbd},2}^{\mathbf{X},\mathbf{Y}}$. This $\mathbf{Z}_2^{\mathrm{an}}$ satisfies Eqs. (10, 11), since $X_2$ and $Y$ are d-separated in $\mathcal{G}_{\mathrm{psbd},2}^{\mathbf{X},\mathbf{Y}}$ given $\mathbf{Z}_2^{\mathrm{an}} \cup \{X_1\}$. Therefore, $\mathbf{Z}^{\mathrm{an}} := (\mathbf{Z}_1^{\mathrm{an}}, \mathbf{Z}_2^{\mathrm{an}}) = (\emptyset, \{Z_a, Z_b, Z_c\})$ satisfies SAC relative to $(\mathbf{X}, \mathbf{Y})$ in $\mathcal{G}$.

### 4.2 Minimal Sequential Adjustment Criterion

We have demonstrated the effectiveness of constructing sequential adjustment sets. However, when dealing with large graphs, $\mathbf{Z}^{\mathrm{an}}$ may include a large number of vertices, leading to high computational costs when evaluating sequential covariate adjustments. This situation highlights the need for a more parsimonious adjustment set. To address this, we introduce the *minimal* sequential covariate adjustment set, which is the smallest subset of $\mathbf{Z}^{\mathrm{an}}$ without sacrificing the validity of the adjustment.

**Definition 8** (Minimal Sequential Covariate Adjustment Set)**.** *Let $(\mathbf{X}, \mathbf{Y})$ denote a disjoint pair in $\mathcal{G}$. An ordered set of vertices $\mathbf{Z}^{\mathrm{min}} := (\mathbf{Z}_1^{\mathrm{min}}, \cdots, \mathbf{Z}_m^{\mathrm{min}})$ where each $\mathbf{Z}_i^{\mathrm{min}}$ are non-descendant of $\mathbf{X}^{\geq i+1}$ is said to be a minimal sequential covariate adjustment set if, for each $i = 1, \cdots, m$,*

1. *$\mathbf{Z}_i^{\mathrm{min}}$ satisfies Eqs. (10, 11).*

2. *For any $\mathbf{Z}_i' \subsetneq \mathbf{Z}_i^{\mathrm{min}}$, $\mathbf{Z}_i'$ does not satisfy Eqs. (10, 11).*

To present the minimal sequential covariate adjustment set, we define a useful tool:

**Definition 9** (Closure (van der Zander and Liśkiewicz, 2020))**.** *For a disjoint vertices $\mathbf{A}, \mathbf{B}$, and $\mathbf{C}$ in $\mathcal{G}$, the closure of $\mathbf{A}$ with respect to $\mathbf{B}$ and $\mathbf{C}$ in $\mathcal{G}$, denoted as $\mathrm{closure}_{\mathcal{G}}(\mathbf{A}; \mathbf{B}, \mathbf{C})$, is the union of $\mathbf{A}$ and the collection of $V \in \mathrm{An}_{\mathcal{G}}(\mathbf{A} \cup \mathbf{B})$ that are connected to $\mathbf{A}$ via a path satisfying the following:*

---

**Algorithm 1:** $\texttt{minSCA}(\mathbf{X}, \mathbf{Y}, \mathcal{G})$

---

**Input:** A disjoint pair of ordered set $(\mathbf{X}, \mathbf{Y})$ and a causal graph $\mathcal{G}$
**Output:** A minimal sequential covariate adjustment set $\mathbf{Z}^{\min}$

1   Set $\mathbf{H}_0^{\mathrm{an}} := \mathbf{Y}_0$.
2   **for** $i = 1, \cdots, m$ **do**
3     Set $\mathbf{Z}_i^{\mathrm{an}}$ as in Eq. (17).
4     $\mathbf{Z}_i^* := \mathrm{closure}_{\mathcal{G}_{\mathrm{psbd},i}^{\mathbf{X},\mathbf{Y}}} (\mathbf{Y}^{\geq i}; X_i, \mathbf{Z}_i^{\mathrm{an}} \cup \mathbf{H}_{i-1}^{\mathrm{an}}) \cap \mathbf{Z}_i^{\mathrm{an}}$
5     $\mathbf{Z}_i^{\min} := \mathrm{closure}_{\mathcal{G}_{\mathrm{psbd},i}^{\mathbf{X},\mathbf{Y}}} (X_i; \mathbf{Y}^{\geq i}, \mathbf{Z}_i^* \cup \mathbf{H}_{i-1}^{\mathrm{an}}) \cap \mathbf{Z}_i^*$
6   **end**
7   **return** $\mathbf{Z}^{\min} := (\mathbf{Z}_1^{\min}, \cdots, \mathbf{Z}_m^{\min})$

---

   1. *The path only contains nodes in* $\mathrm{An}_{\mathcal{G}}(\mathbf{A} \cup \mathbf{B})$*; and*

   2. *All non-collider vertices on the path are not in* $\mathbf{C}$*; Equivalently, any vertex on the path that is in* $\mathbf{C}$ *is a collider on the path.*

Verbally speaking, $\mathrm{closure}_{\mathcal{G}}(\mathbf{A}; \mathbf{B}, \mathbf{C})$ is a set of vertices in $\mathrm{An}_{\mathcal{G}}(\mathbf{A} \cup \mathbf{B})$ that is d-connected to $\mathbf{A}$ in $\mathcal{G}$ conditioned on $\mathbf{C}$, where the paths are through $\mathrm{An}_{\mathcal{G}}(\mathbf{A} \cup \mathbf{B})$. Finding the closure can be efficiently done in $O(|\mathbf{V}| + |\mathbf{E}|)$ time (van der Zander and Liśkiewicz, 2020), where $|\mathbf{V}|$ and $|\mathbf{E}|$ are the number of vertices and edges of the graph $\mathcal{G}$. For example, consider $\mathrm{closure}_{\mathcal{G}_{\mathrm{psbd},2}^{\mathbf{X},\mathbf{Y}}} (X_2; Y, \{Z_a, Z_b, Z_c, X_1\})$ with $\mathcal{G}_{\mathrm{psbd},2}^{\mathbf{X},\mathbf{Y}}$ in Fig. 4c. First, the ancestor of $\{X_2, Y\}$ are $\{Z_a, Z_b, Z_c, X_1\}$. Since $Z_a$ and $X_1$ are adjacent to $X_2$, they are contained in the closure. However, $\{Z_b, Z_c\}$ are not in the closure, since all the non-collider vertices on the path between $X_2$ and $\{Z_b, Z_c\}$ are contained in $\{Z_a, Z_b, Z_c, X_1\}$. Therefore, $\{Z_a, X_1, X_2\} = \mathrm{closure}_{\mathcal{G}_{\mathrm{psbd},2}^{\mathbf{X},\mathbf{Y}}} (X_2; Y, \{Z_a, Z_b, Z_c, X_1\})$.

Equipped with the closure, we propose a procedure to construct the minimal sequential covariate adjustment in Algo. 1. The minimal sequential covariate adjustment set, $\mathbf{Z}^{\min}$, is derived directly from the ordered set $\mathbf{Z}^{\mathrm{an}}$, which characterizes the sequential covariate adjustment. Therefore, $\mathbf{Z}^{\min}$ naturally retains this characterizing property.

**Theorem 4** (Construction of Minimal Sequential Covariate Adjustment Set)**.** *Let* $(\mathbf{X}, \mathbf{Y})$ *denote a disjoint pair in* $\mathcal{G}$*. Then, the following statements are equivalent:*

   1. *There exists an ordered set* $\mathbf{Z} := (\mathbf{Z}_1, \cdots, \mathbf{Z}_m)$ *satisfying SAC w.r.t.* $(\mathbf{X}, \mathbf{Y})$ *in* $\mathcal{G}$*.*

   2. *The ordered set* $\mathbf{Z}^{\min} = \texttt{minSCA}(\mathbf{X}, \mathbf{Y}, \mathcal{G})$ *from Algo. 1 is a minimal sequential covariate adjustment set.*

Constructing $\mathbf{Z}^{\min} = \texttt{minSCA}(\mathbf{X}, \mathbf{Y}, \mathcal{G})$ can be achieved in $O(m(|\mathbf{V}| + |\mathbf{E}|))$, as the construction of each $\mathbf{Z}_i^{\mathrm{an}}$ in Eq. (17) and the required closures each take $O(|\mathbf{V}| + |\mathbf{E}|)$ and are repeated $m$ times.

We now demonstrate Thm. 4 with a causal diagram $\mathcal{G}$ in Fig. 4a. We have shown that $\mathbf{Z}^{\mathrm{an}} := (\mathbf{Z}_1^{\mathrm{an}}, \mathbf{Z}_2^{\mathrm{an}})$ with $\mathbf{Z}_1^{\mathrm{an}} := \emptyset$ and $\mathbf{Z}_2^{\mathrm{an}} := \{Z_a, Z_b, Z_c\}$ satisfies SAC relative to $(\mathbf{X}, Y)$ in $\mathcal{G}$. Therefore, we only demonstrate Algo. 1 for $\mathbf{Z}_2^{\mathrm{an}} := \{Z_a, Z_b, Z_c\}$. Run line 4: $\mathbf{Z}_2^* := \mathrm{closure}_{\mathcal{G}_{\mathrm{psbd},2}^{\mathbf{X},\mathbf{Y}}} (Y; X_2, \{Z_a, Z_b, Z_c, X_1\}) \cap \{Z_a, Z_b, Z_c\}$. First, $\{Z_a, Z_c\}$ is included in the closure since they are adjacent to $Y$. Next, $Z_b$ is included in the closure, since the path $Y \leftrightarrow Z_c \leftarrow Z_b$ does not include non-collider vertex. Therefore, $\mathbf{Z}_2^* = \mathbf{Z}_2^{\mathrm{an}} = \{Z_a, Z_b, Z_c\}$. Next, run line 5. We already witnessed that $\{Z_a, X_1, X_2\} = \mathrm{closure}_{\mathcal{G}_{\mathrm{psbd},2}^{\mathbf{X},\mathbf{Y}}} (X_2; Y, \{Z_a, Z_b, Z_c, X_1\})$, which leads to $\mathbf{Z}_2^{\min} := \{Z_a, X_1, X_2\} \cap \{Z_a, Z_b, Z_c\} = \{Z_a\}$. Therefore, $\mathbf{Z}^{\min} := (\mathbf{Z}_1^{\min}, \mathbf{Z}_2^{\min}) = (\emptyset, \{Z_a\})$.

## 5   Conclusion

We develop a sound and complete graphical criterion for sequential covariate adjustment. We start by highlighting the limitations of the existing mSBD criterion, acknowledging its soundness but noting its lack of completeness. (Prop. 3). Against this background, we introduce the *sequential adjustment criterion* (Def. 7), which characterizes sequential covariate adjustment (Thm. 1). The

proposed criterion includes and extends the mSBD and standard adjustment criterion, confirming its comprehensive nature since they imply the sequential adjustment criterion (Prop. 1 and Thm. 2). Further, we develop a procedure for constructing the sequential adjustment set that captures the sequential covariate adjustment (Thm. 3). Finally, we devise an algorithm to identify the minimal sequential covariate adjustment set (Algo. 1 and Thm. 4).

## Acknowledgments and Disclosure of Funding

We thank anonymous reviewers for constructive comments to improve the manuscript. This work was partly supported by the IITP (RS-2022-II220953/20%), NRF (RS-2023-00211904/20%, RS-2023-00222663/20%), and MFDS (23212MFDS202/20%) grant funded by the Korean government.

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

# Supplement to "Complete Graphical Criterion for Sequential Covariate Adjustment in Causal Inference"

## Contents

# A Proofs

In this appendix, we provide detailed proof of Theorems presented in the main paper.

**Notations.** We symbolize the d-connection between two vertices $A$ and $B$ as $A \!-\! B$. Furthermore, we symbolize the type of paths, that are not mutually exclusive, as follows:

1. $A \multimap\!\!\rightarrow B$ means that there is a directed path from $A$ to $B$.

2. $A \leftarrow\!\!\circ\!\!\rightarrow B$ means that there is a non-directed path with no colliders and all non-colliders on the path are not conditioned.

3. $A \ast\!\!\rightarrow\!\!\bullet\!\!\leftarrow\!\!\ast B$ means that there is a non-directed path containing at least one collider, and either the collider or one of its descendants is conditioned.

## A.1 Proof of Theorem 1

In order to prove both the soundness and completeness of SAC, we structure the proof in two parts. First, we will demonstrate the soundness of SAC through Lemma S.1, followed by proving its completeness in Lemma S.2.

**Lemma S.1** (Soundness of SAC). *Let $(\mathbf{X}, \mathbf{Y})$ denote a disjoint pair of ordered sets where $\mathbf{Y}$ is partitioned with $\mathrm{PT}_{\mathcal{G}}^{\mathbf{X}}$. Let $\mathbf{Z} := (\mathbf{Z}_1, \cdots, \mathbf{Z}_m)$ denote an ordered set of vertices disjoint to $(\mathbf{X}, \mathbf{Y})$ in $\mathcal{G}$. If $\mathbf{Z}$ satisfies the SAC relative to $(\mathbf{X}, \mathbf{Y})$ in $\mathcal{G}$, then $\mathbf{Z}$ is a sequential adjustment set relative to $(\mathbf{X}, \mathbf{Y})$ in $\mathcal{G}$.*

***Proof of Lemma S.1.*** We first assume that the following claim holds, and then demonstrate that the claim is valid. For any fixed $\mathbf{Z}_i \in (\mathbf{Z}_1, \cdots, \mathbf{Z}_m)$, suppose the following claim is *true*.

---

**Claim.** If $\mathbf{Z}$ satisfies SAC relative to $(\mathbf{X}, \mathbf{Y})$ in $\mathcal{G}$, then there exists an ordered set $\mathbf{Z}_i = (Z_{i,1}, \cdots, Z_{i,m_i})$ (where $m_i = |\mathbf{Z}_i|$) such that, for each $Z_{i,j}$ in the ordered set, and with $\mathbf{Z}_i^{(j-1)} := (Z_{i,1}, \cdots, Z_{i,j-1})$, at least one of the following two cases holds:

$$\text{Case 1. } (\mathbf{Y}^{\geq i} \perp\!\!\!\perp Z_{i,j} \mid \mathbf{X}^{\geq i}, \mathbf{Z}_i^{(j-1)}, \mathbf{H}_{i-1})_{\mathcal{G}_{\overline{\mathbf{X}^{\geq i}}}}.$$

$$\text{Case 2. } (F_{X_i} \perp\!\!\!\perp Z_{i,j} \mid \mathbf{Z}_i^{(j-1)}, \mathbf{H}_{i-1})_{\mathcal{G}}.$$

where $F_{X_i}$ is a new parent of $X_i$, called the *regime*, augmented in the graph $\mathcal{G}$.

---

For the fixed partition of $\mathbf{Z}_i$ that makes the claim true, suppose Case 1 holds for all $Z_{i,j} \in \mathbf{Z}_i$. Then,

$$P(\mathbf{y}^{\geq i} \mid \mathrm{do}(\mathbf{x}^{\geq i}), \mathbf{z}_i^{(j-1)}, \mathbf{h}_{i-1})$$

$$= \sum_{z_{i,j}} P(\mathbf{y}^{\geq i} \mid \mathrm{do}(\mathbf{x}^{\geq i}), \mathbf{z}_i^{(j-1)}, \mathbf{h}_{i-1}) P(z_{i,j} \mid \mathbf{z}_i^{(j-1)}, \mathbf{h}_{i-1})$$

$$= \sum_{z_{i,j}} P(\mathbf{y}^{\geq i} \mid \mathrm{do}(\mathbf{x}^{\geq i}), \mathbf{z}_i^{(j)}, \mathbf{h}_{i-1}) P(z_{i,j} \mid \mathbf{z}_i^{(j-1)}, \mathbf{h}_{i-1}) \qquad \text{by Case 1.}$$

Now, suppose Case 2 holds for all $Z_{i,j}$. Then,

$$P(\mathbf{y}^{\geq i} \mid \mathrm{do}(\mathbf{x}^{\geq i}), \mathbf{z}_i^{(j-1)}, \mathbf{h}_{i-1})$$

$$= \sum_{z_{i,j}} P(\mathbf{y}^{\geq i} \mid \mathrm{do}(\mathbf{x}^{\geq i}), \mathbf{z}_i^{(j)}, \mathbf{h}_{i-1}) P(z_{i,j} \mid \mathrm{do}(\mathbf{x}^{\geq i}), \mathbf{z}_i^{(j-1)}, \mathbf{h}_{i-1})$$

$$= \sum_{z_{i,j}} P(\mathbf{y}^{\geq i} \mid \mathrm{do}(\mathbf{x}^{\geq i}), \mathbf{z}_i^{(j)}, \mathbf{h}_{i-1}) P(z_{i,j} \mid \mathrm{do}(x_i), \mathbf{z}_i^{(j-1)}, \mathbf{h}_{i-1}) \qquad \text{by the assumption on } \mathbf{Z}_i$$

$$= \sum_{z_{i,j}} P(\mathbf{y}^{\geq i} \mid \mathrm{do}(\mathbf{x}^{\geq i}), \mathbf{z}_i^{(j)}, \mathbf{h}_{i-1}) P(z_{i,j} \mid \mathbf{z}_i^{(j-1)}, \mathbf{h}_{i-1}) \qquad \text{by Case 2.}$$

In summary, under the claim, for all $Z_{i,j} \in \mathbf{Z}_i$,

$$P(\mathbf{y}^{\geq i} \mid \mathrm{do}(\mathbf{x}^{\geq i}), \mathbf{z}_i^{(j-1)}, \mathbf{h}_{i-1}) = \sum_{z_{i,j}} P(\mathbf{y}^{\geq i} \mid \mathrm{do}(\mathbf{x}^{\geq i}), \mathbf{z}_i^{(j)}, \mathbf{h}_{i-1}) P(z_{i,j} \mid \mathbf{z}_i^{(j-1)}, \mathbf{h}_{i-1}).$$

By applying this equation for all elements $Z_{i,1}, \cdots, Z_{i,m_i}$ in $\mathbf{Z}_i$, we have

$$P(\mathbf{y}^{\geq i} \mid \mathrm{do}(\mathbf{x}^{\geq i}), \mathbf{h}_{i-1}) = \sum_{\mathbf{z}_i} P(\mathbf{y}^{\geq i} \mid \mathrm{do}(\mathbf{x}^{\geq i}), \mathbf{z}_i, \mathbf{h}_{i-1}) P(\mathbf{z}_i \mid \mathbf{h}_{i-1}). \tag{A.1}$$

We also note that the given condition in Eq. (10) implies the Rule 2 of the do-calculus (Pearl, 2000):

$$(\mathbf{Y}^{\geq i} \perp\!\!\!\perp X_i \mid \mathbf{Z}_i, \mathbf{H}_{i-1})_{\mathcal{G}_{\mathrm{psbd},i}^{\mathbf{X},\mathbf{Y}}} \implies (\mathbf{Y}^{\geq i} \perp\!\!\!\perp X_i \mid \mathbf{Z}_i, \mathbf{H}_{i-1})_{\mathcal{G}_{\underline{X_i}\overline{\mathbf{x}^{\geq i+1}}}} \tag{A.2}$$

since the d-separation is always preserved when cutting the edge from the graph. The right-hand side of Eq. (A.1) can be written further as follows:

$$\sum_{\mathbf{z}_i} P(\mathbf{y}^{\geq i} \mid \mathrm{do}(\mathbf{x}^{\geq i}), \mathbf{h}_{i-1}, \mathbf{z}_i) P(\mathbf{z}_i \mid \mathbf{h}_{i-1})$$

$$= \sum_{\mathbf{z}_i} P(\mathbf{y}^{\geq i} \mid x_i, \mathrm{do}(\mathbf{x}^{\geq i+1}), \mathbf{h}_{i-1}, \mathbf{z}_i) P(\mathbf{z}_i \mid \mathbf{h}_{i-1}) \qquad \text{by Eq. (A.2)}$$

$$= \sum_{\mathbf{z}_i} P(\mathbf{y}^{\geq i+1} \mid \mathrm{do}(\mathbf{x}^{\geq i+1}), \mathbf{h}_i) P(\mathbf{y}_i \mid \mathrm{do}(\mathbf{x}^{\geq i+1}), \mathbf{h}_{i-1}, x_i, \mathbf{z}_i) P(\mathbf{z}_i \mid \mathbf{h}_{i-1})$$

$$= \sum_{\mathbf{z}_i} P(\mathbf{y}^{\geq i+1} \mid \mathrm{do}(\mathbf{x}^{\geq i+1}), \mathbf{h}_i) P(\mathbf{y}_i \mid \mathbf{h}_{i-1}, x_i, \mathbf{z}_i) P(\mathbf{z}_i \mid \mathbf{h}_{i-1}) \qquad \text{by } \mathrm{PT}_{\mathcal{G}}^{\mathbf{X}}.$$

Therefore,

$$P(\mathbf{y} \mid \mathrm{do}(\mathbf{x}))$$

$$= P(\mathbf{y}^{\geq 1} \mid \mathrm{do}(\mathbf{x}^{\geq 1}), \mathbf{h}_0) P(\mathbf{h}_0)$$

$$= \sum_{\mathbf{z}_1} P(\mathbf{y}^{\geq 2} \mid \mathrm{do}(\mathbf{x}^{\geq 2}), \mathbf{h}_1) P(\mathbf{y}_1 \mid \mathbf{h}_1, x_1, \mathbf{z}_1) P(\mathbf{z}_1 \mid \mathbf{h}_0) P(\mathbf{h}_0)$$

$$= \sum_{\mathbf{z}_1, \mathbf{z}_2} P(\mathbf{y}^{\geq 3} \mid \mathrm{do}(\mathbf{x}^{\geq 3}), \mathbf{h}_2) P(\mathbf{y}_2 \mid \mathbf{h}_1, x_2, \mathbf{z}_2) P(\mathbf{z}_2 \mid \mathbf{h}_1) P(\mathbf{y}_1 \mid \mathbf{z}_1, x_1, \mathbf{z}_1) P(\mathbf{z}_1 \mid \mathbf{h}_0) P(\mathbf{h}_0)$$

$$\vdots$$

$$= \sum_{\mathbf{z}} \prod_{i=0}^{m} P(\mathbf{y}_i \mid \mathbf{h}_{i-1}, x_i, \mathbf{z}_i) \prod_{i=j}^{m} P(\mathbf{z}_j \mid \mathbf{h}_{j-1})$$

$$= \sum_{\mathbf{z}} \prod_{i=0}^{m} P(\mathbf{y}_i \mid \mathbf{h}_{i-1}, x_i, \mathbf{z}_i) P(\mathbf{z}_{i+1} \mid \mathbf{h}_i)$$

$$= \sum_{\mathbf{z}} \prod_{i=0}^{m} P(\mathbf{z}_{i+1}, \mathbf{y}_i \mid \mathbf{h}_{i-1}, x_i, \mathbf{z}_i).$$

In conclusion, the sequential adjustment criterion is *sound* under the claim. It now remains to prove that the claim is true.

***Proof of the claim.*** We will prove the claim by contradiction. Suppose that $\mathbf{Z}$ satisfies the SAC relative to $(\mathbf{X}, \mathbf{Y})$ in $\mathcal{G}$ but for all orders of elements in $\mathbf{Z}_i$ (denoting an order as $\pi(\mathbf{Z}_i)$), there always exists some $Z_{i,j} \in \mathbf{Z}_i$ with $\mathbf{Z}_i^{(j-1)} := \mathbf{Z}_i^{(\pi, j-1)}$ (which is predecessors of $\mathbf{Z}_{i,j}$ under the order $\pi(\mathbf{Z}_i)$) such that $Z_{i,j}$ does not satisfy either Case 1 or Case 2, i.e.,

$$(\mathbf{Y}^{\geq i} \not\perp\!\!\!\perp Z_{i,j} \mid \mathbf{X}^{\geq i}, \mathbf{Z}_i^{(j-1)}, \mathbf{H}_{i-1})_{\mathcal{G}_{\overline{\mathbf{x}^{\geq i}}}} \text{ and } (F_{X_i} \not\perp\!\!\!\perp Z_{i,j} \mid \mathbf{Z}_i^{(j-1)}, \mathbf{H}_{i-1})_{\mathcal{G}}. \tag{A.3}$$

This means that there exists a d-connecting path between $Z_{i,j}$ and $Y \in \mathbf{Y}^{\geq i}$ in $\mathcal{G}_{\overline{\mathbf{X}^{\geq i}}}$ given $\mathbf{X}^{\geq i} \cup \mathbf{Z}_i^{(j-1)} \cup \mathbf{H}_{i-1}$, and a d-connecting path between $F_{X_i}$ and $Z_{i,j}$ in $\mathcal{G}$ given $\mathbf{Z}_i^{(j-1)} \cup \mathbf{H}_{i-1}$. That is, the following path exists:

$$(F_{X_i} \to X_i - Z_{i,j})_{\mathcal{G}} \wedge (Z_{i,j} - Y)_{\mathcal{G}_{\overline{\mathbf{X}^{\geq i}}}}. \tag{A.4}$$

We further state the following. In this setting, there exists a d-connecting path between $Z_{i,j}$ and $Y$ in $\mathcal{G}_{\overline{\mathbf{X}^{\geq i+1}}}$, which is evident since adding an edge (from $\mathcal{G}_{\overline{\mathbf{X}^{\geq i}}}$ to $\mathcal{G}_{\overline{\mathbf{X}^{\geq i+1}}}$) does not block any d-connecting paths. Moreover, the d-connecting path between $F_{X_i}$ and $Z_{i,j}$ in $\mathcal{G}$ is preserved in $\mathcal{G}_{\overline{\mathbf{X}^{\geq i+1}}}$. To witness, suppose there exists a d-connected path between $F_{X_i}$ and $Z_{i,j}$ in $\mathcal{G}$, while the path is blocked/cut in $\mathcal{G}_{\overline{\mathbf{X}^{\geq i+1}}}$. This implies that the path contains an edge $\to X_k$ for some $X_k \in \mathbf{X}^{\geq i+1}$. By the topological order and the assumption that $\mathbf{Z}_i$ is non-descendant of $\mathbf{X}^{\geq i+1}$, the directed path from $X_k$ to $X_i$ and the directed path from $X_k$ to $Z_{i,j}$ are both impossible. Thus, we only consider non-directed path from $F_{X_i}$ to $Z_{i,j}$. First, $X_k$ on the path connecting $X_i$ and $Z_{i,j}$ cannot be a collider, as the path would be d-separated if $X_k$ were not conditioned. Additionally, $X_k$ cannot be an ancestor of $\mathbf{Z}_i^{(j-1)} \cup \mathbf{H}_{i-1}$ due to the topological order of $\mathbf{X}$, the mechanism of the partitioning operator $\mathrm{PT}_{\mathcal{G}}^{\mathbf{X}}$, and the assumption that $\mathbf{Z}_j$ is non-descendant of $\mathbf{X}^{\geq j+1}$. Therefore, we consider the path between $X_i$ and $Z_{i,j}$ through $X_k$, where neither $X_k$ nor its descendant forms a collider. The possible types of paths in $\mathcal{G}$ are as follows:

Type 1. $X_i \mathrel{-\!\circ\!\to} X_k \to\bullet\leftarrow\!\!* Z_{i,j}$
Type 2. $X_i \leftarrow\!\circ\!\to X_k \to\bullet\leftarrow\!\!* Z_{i,j}$
Type 3. $X_i *\!\!\to\bullet\leftarrow X_k \leftarrow\!\circ\!- Z_{i,j}$
Type 4. $X_i *\!\!\to\bullet\leftarrow X_k \leftarrow\!\circ\!\to Z_{i,j}$
Type 5. $X_i *\!\!\to\bullet\leftarrow X_k *\!\!\to\bullet\leftarrow\!\!* Z_{i,j}$
Type 6. $X_i *\!\!\to\bullet\leftarrow\!\!* X_k \to\bullet\leftarrow\!\!* Z_{i,j}$

All cases imply the existence of a directed path from $X_k$ to any vertex in $\mathbf{Z}_i^{(j-1)} \cup \mathbf{H}_{i-1}$, making $X_k$ an ancestor of $\mathbf{Z}_i^{(j-1)} \cup \mathbf{H}_{i-1}$. This contradicts our setting, just as in the case where $X_k$ is a collider. It confirms that the d-connecting path between $F_{X_i}$ and $Z_{i,j}$ in $\mathcal{G}$ is preserved in $\mathcal{G}_{\overline{\mathbf{X}^{\geq i+1}}}$. As a result, Eq. (A.4) means the presence of the following path in $\mathcal{G}_{\overline{\mathbf{X}^{\geq i+1}}}$:

$$(F_{X_i} \to X_i - Z_{i,j} - Y)_{\mathcal{G}_{\overline{\mathbf{X}^{\geq i+1}}}}.$$

Then, there are nine types of d-connected path between $X_i$ and $Y$ in $\mathcal{G}_{\overline{\mathbf{X}^{\geq i+1}}}$ given $\mathbf{Z}_i^{(j-1)} \cup \mathbf{H}_{i-1}$:

Type 1. $F_{X_i} \to X_i \mathrel{-\!\circ\!\to} Z_{i,j} \mathrel{-\!\circ\!\to} Y$
Type 2. $F_{X_i} \to X_i \mathrel{-\!\circ\!\to} Z_{i,j} \leftarrow\!\circ\!\to Y$
Type 3. $F_{X_i} \to X_i \mathrel{-\!\circ\!\to} Z_{i,j} *\!\!\to\bullet\leftarrow\!\!* Y$
Type 4. $F_{X_i} \to X_i \leftarrow\!\circ\!\to Z_{i,j} \mathrel{-\!\circ\!\to} Y$
Type 5. $F_{X_i} \to X_i \leftarrow\!\circ\!\to Z_{i,j} \leftarrow\!\circ\!\to Y$
Type 6. $F_{X_i} \to X_i \leftarrow\!\circ\!\to Z_{i,j} *\!\!\to\bullet\leftarrow\!\!* Y$
Type 7. $F_{X_i} \to X_i *\!\!\to\bullet\leftarrow\!\!* Z_{i,j} \mathrel{-\!\circ\!\to} Y$
Type 8. $F_{X_i} \to X_i *\!\!\to\bullet\leftarrow\!\!* Z_{i,j} \leftarrow\!\circ\!\to Y$
Type 9. $F_{X_i} \to X_i *\!\!\to\bullet\leftarrow\!\!* Z_{i,j} *\!\!\to\bullet\leftarrow\!\!* Y$

We will show that each type of path belongs to one of the following three cases:

Case 1. There always exists another order $\pi'$ for $\mathbf{Z}_i$ such that the path is blocked, even if the path is d-connected under the oreder $\pi$.

Case 2. The path contradicts the first condition of SAC in Eq. (10).

Case 3. The path contradicts the second condition of SAC in Eq. (11).

The paths in Type 1 are directed from $X_i$ to $Y$, which contradicts the second condition of SAC in Eq. (11).

Consider Type 2. Since $Z_{i,j}$ is a collider, and $\mathbf{Z}_i^{(j-1)} \cup \mathbf{H}_{i-1}$ is conditioned on, it forms a d-separated path between $X_i$ and $Y$, which is preserved in $\mathcal{G}_{\mathrm{psbd},i}^{\mathbf{X},\mathbf{Y}}$. First, consider the case where this path is not blocked by further conditioning on $\mathbf{Z}_i \setminus \mathbf{Z}_i^{(j-1)}$. This implies that Eq. (10) is not satisfied, thereby, contradicting the initial assumption that $\mathbf{Z}$ satisfies SAC. Therefore, the path must be blocked by further conditioning on $\mathbf{Z}_i \setminus \mathbf{Z}_i^{(j-1)}$. In other words, there exists another order $\pi'$ for $\mathbf{Z}_i$ such that this path is blocked whenever $\{Z_{i,j}\} \cup \mathbf{Z}_i^{(\pi',j-1)} \cup \mathbf{H}_{i-1}$ is conditioned on. Therefore, without loss of generality, we can ignore Type 2 by considering any other order $\pi'$ in which the colliding path is blocked.

Consider Type 3. Since the case where $Z_{i,j}$ forms a collider is considered in Type 2, we only need to consider the case where $Z_{i,j} \to$ lies on the path. In this case, $Z_{i,j}$ has a directed path to any vertex in $\mathbf{H}_{i-1} \cup \mathbf{Z}_i^{(j-1)}$. However, it is impossible for the vertex to be involved in $\mathbf{H}_{i-1}$, as this would contradict the valid topological order of SAC. Therefore, we can conclude that there exists a directed path from $X_i$ to $Z \in \mathbf{Z}_i^{(j-1)}$ through $Z_{i,j}$. Then, the following two cases are possible: First, there exists a directed path from $Y$ to $Z$. This case contradicts the condition of SAC in Eq. (11), as $Z$ would be a descendant of $Y$. Second, there exists a divergent path between $Y$ and $Z$, meaning that $Z$ forms a collider on the path between $X_i$ and $Y$. Note that we can ignore Type 3 for the same reason discussed in the case of Type 2.

The path for Type 4-6 be open when $X_i$ is an ancestor of the conditioning set $\mathbf{Z}_i^{(j-1)} \cup \mathbf{H}_{i-1}$. Due to the valid topological order of SAC, we only consider the case where the conditioning set is $\mathbf{Z}_i^{(j-1)}$. This implies the existence of some path of the form $F_{X_i} \to X_i \to Z\ (\in \mathbf{Z}_i^{(j-1)})$ that does not belong to Type 4-6, allowing us to reduce this types to another type.

We now consider the path for Type 7-9. As we witnessed that $X_i$ cannot be a collider opening the path, it should contain $X_i \to$, which implies that $X_i$ is an ancestor of some collider and its descendant is some vertices in $\mathbf{Z}_i^{(j-1)} \cup \mathbf{H}_{i-1}$. Hence, the collider must be a vertex in $\mathbf{Z}_i^{(j-1)}$, as $\mathbf{H}_{i-1}$ being the collider would contradicts the valid topological order of SAC. Therefore, we can ignore such type of paths.

Therefore, any type of path leads to a contradiction, which validates the claim. In conclusion, there exists an order such that every $Z_{i,j} \in \mathbf{Z}_i$ satisfies Case 1 or Case 2, meaning the claim is *true*. $\qquad\square$

**Lemma S.2** (Completeness of SAC). *Let $(\mathbf{X}, \mathbf{Y})$ denote a disjoint pair of ordered sets where $\mathbf{Y}$ is partitioned with $\mathrm{PT}_{\mathcal{G}}^{\mathbf{X}}$. Let $\mathbf{Z} := (\mathbf{Z}_1, \cdots, \mathbf{Z}_m)$ denote an ordered set of vertices disjoint to $(\mathbf{X}, \mathbf{Y})$. There exists a graph $\mathcal{G}$ such that $\mathbf{Z}$ is not a sequential adjustment set relative to $(\mathbf{X}, \mathbf{Y})$ in $\mathcal{G}$ whenever $\mathbf{Z}$ does not satisfy SAC relative to $(\mathbf{X}, \mathbf{Y})$ in $\mathcal{G}$.*

***Proof of Lemma S.2.*** If $\mathbf{Z}$ does not satisfy SAC in $\mathcal{G}$, then there exists $\mathbf{Z}_i \in \mathbf{Z}$ not satisfying Eq. (10) or Eq. (11) in SAC.

First, suppose $\mathbf{Z}_i$ does not satisfy Eq. (10). Then, there must be a path between $X_i$ and $Y \in \mathbf{Y}^{\geq i}$ such that all non-colliders on the path are not in $\mathbf{Z}_i \cup \mathbf{H}_{i-1}$ and all colliders on the path are in $\mathrm{An}_{\mathcal{G}_{\mathrm{psbd},i}^{\mathbf{X},\mathbf{Y}}}(\mathbf{Z}_i \cup \mathbf{H}_{i-1})$. Since the path is defined in $\mathcal{G}_{\mathrm{psbd},i}^{\mathbf{X},\mathbf{Y}}$ and by the mechanism of the partitioning operator $\mathrm{PT}_{\mathcal{G}}^{\mathbf{X}}$, the path cannot be a directed path.

Now, suppose the path is a divergent path that does not contain any colliders; i.e., $X_i \leftarrow\!\circ\!\rightarrow Y$. In this case, we can consider a simple graph $\mathcal{G} = \{X_i \leftrightarrow Y_i, X_i \to Y_i\}$ where $\mathcal{G}_{\mathrm{psbd},i}^{\mathbf{X},\mathbf{Y}}$ contains a subgraph $\{X_i \leftrightarrow Y_i\}$. Then, $P(\mathbf{y} \mid \mathrm{do}(\mathbf{x}))$ cannot be given as a sequential adjustment since it is not even identifiable. Suppose, instead, the path is a colliding path; i.e., $X \ast\!\!\to\!\bullet\!\leftarrow\!\!\ast Y$ where the collider is an ancestor of the conditioned vertices $\mathbf{Z}_i \cup \mathbf{H}_{i-1}$. Then, we can consider a simple graph $\mathcal{G}$ that contains a subgraph $\{X_i \leftarrow Z_i \to Y_i, X_i \to Y_i, X_i \leftrightarrow Z_i \leftrightarrow Y_i\}$. Here again, $P(\mathbf{y} \mid \mathrm{do}(\mathbf{x}))$ cannot be given as a sequential adjustment since it is not even identifiable.

Next, suppose $\mathbf{Z}_i$ does not satisfy Eq. (11). Then, we may consider a graph a simple graph $\mathcal{G}$ that contains a subgraph $X_i \rightarrow Z_i \rightarrow Y_i$. Then, there exists a SCM $\mathcal{M}$ such that $P(\mathbf{y} \mid \mathrm{do}(\mathbf{x}))$ is not given as an adjustment over $Z_i$ (Shpitser et al., 2010).

This concludes the proof that the sequential adjustment criterion is *complete*. □

## A.2 Proof of Corollary 1

We will prove that if there exists $\mathbf{Z}_i \in \mathbf{Z}$ that does not satisfy Eq. (10) or Eq. (11) in SAC, then $\mathbf{Z}_i$ does not satisfy Eq. (4) or Eq. (5) in mSBD criterion.

For the sake of contradiction, suppose $\mathbf{Z}_i$ does not satisfy Eq. (10), but $(X_i \perp\!\!\!\perp \mathbf{Y}^{\geq i} \mid \mathbf{H}_{i-1}, \mathbf{Z}_i)_{\mathcal{G}_{\underline{X_i}\overline{\mathbf{X}^{\geq i+1}}}}$ holds. This means that there exists a d-connecting path between $X_i$ and $Y \in \mathbf{Y}^{\geq i}$ given $\mathbf{Z}_i \cup \mathbf{H}_{i-1}$ in $\mathcal{G}_{\mathrm{psbd},i}^{\mathbf{X},\mathbf{Y}}$ while the path is blocked in $\mathcal{G}_{\underline{X_i}\overline{\mathbf{X}^{\geq i+1}}}$. The differences between $\mathcal{G}_{\underline{X_i}\overline{\mathbf{X}^{\geq i+1}}}$ and $\mathcal{G}_{\mathrm{psbd},i}^{\mathbf{X},\mathbf{Y}}$ imply that the d-connecting path should contain $X_i \rightarrow$. Since the path cannot be a directed path, it must be a colliding path where the collider is an ancestor of any vertices in $\mathbf{Z}_i \cup \mathbf{H}_{i-1}$. This means that there exists a directed path either $X_i \multimap\rightarrow Z$ for some $Z \in \mathbf{Z}_i$ or $X_i \multimap\rightarrow H$ for some $H \in \mathbf{H}_{i-1}$. If $X_i \multimap\rightarrow Z$, such $\mathbf{Z}_i$ fails to satisfy the mSBD criterion because $\mathbf{Z}_i$ contains a descendant of $X_i$. If $X_i \multimap\rightarrow H$, it implies that there exists a directed path from $X_i$ to $X_k \in \mathbf{X}^{(i-1)}$, $Y_k \in \mathbf{Y}^{(i-1)}$ or $Z_k \in \mathbf{Z}^{(i-1)}$. In any case, it contradicts the topological order of $\mathbf{X}$, the mechanism of partition operator $\mathrm{PT}_{\mathcal{G}}^{\mathbf{X}}$, or the assumption that $\mathbf{Z}_{i-1}$ is non-descendant of $\mathbf{X}^{\geq i}$ in $\mathcal{G}$.

Now, suppose $\mathbf{Z}_i$ does not satisfy Eq. (11). Then, such $\mathbf{Z}_i$ does not satisfy Eq. (4) since failure to satisfy Eq. (11) means that $\mathbf{Z}_i$ is a descendant of $\mathbf{X}_i$ in $\mathcal{G}$.

This concludes the proof that the mSBD criterion implies the SAC.

## A.3 Proof of Theorem 2

We will prove that if the SAC fails for any partition of $\mathbf{Z}$ across all topological ordering of $\mathbf{X}$, then $\mathbf{Z}$ also fails to satisfy the adjustment criterion. The failure of SAC at $\mathbf{Z}_i$ can be stated as

$$(\mathbf{Y}^{\geq i} \not\perp\!\!\!\perp X_i \mid \mathbf{Z}_i \cup \mathbf{H}_{i-1})_{(\mathcal{G}_{\overline{\mathbf{X}^{\geq i+1}}})_{\mathrm{pbd}}^{X_i,\mathbf{Y}^{\geq i}}}, \text{ for any } \mathbf{Z}_i \subseteq \mathbf{Z} \setminus \mathrm{De}_{\mathcal{G}}(\mathbf{X}^{\geq i+1}). \quad (A.5)$$

Then, our goal is to show the following:

---

**Statement to be proved:** For any $\mathbf{Z}_i \subseteq \mathbf{Z} \setminus \mathrm{De}_{\mathcal{G}}(\mathbf{X}^{\geq i+1})$,

$$(\mathbf{Y}^{\geq i} \not\perp\!\!\!\perp X_i \mid \mathbf{Z}_i \cup \mathbf{H}_{i-1})_{(\mathcal{G}_{\overline{\mathbf{X}^{\geq i+1}}})_{\mathrm{pbd}}^{X_i,\mathbf{Y}^{\geq i}}}$$
$$\implies (\mathbf{Y}^{\geq i} \not\perp\!\!\!\perp X_i \mid \mathbf{Z} \cup \mathbf{H}_{i-1})_{\mathcal{G}_{\mathrm{pbd}}^{\mathbf{X},\mathbf{Y}}}. \quad (A.6)$$

---

Note that the right-hand side of Eq. (A.6) implies $(\mathbf{Y} \not\perp\!\!\!\perp \mathbf{X} \mid \mathbf{Z})_{\mathcal{G}_{\mathrm{pbd}}^{\mathbf{X},\mathbf{Y}}}$, meaning that $\mathbf{Z}$ does not satisfy the adjustment criterion relative to $(\mathbf{X}, \mathbf{Y})$ in $\mathcal{G}$. To this end, we first witness the following:

---

**Step 1.** For any $\mathbf{Z}_i \subseteq \mathbf{Z} \setminus \mathrm{De}_{\mathcal{G}}(\mathbf{X}^{\geq i+1})$,

$$(\mathbf{Y}^{\geq i} \not\perp\!\!\!\perp X_i \mid \mathbf{Z}_i \cup \mathbf{H}_{i-1})_{(\mathcal{G}_{\overline{\mathbf{X}^{\geq i+1}}})_{\mathrm{pbd}}^{X_i,\mathbf{Y}^{\geq i}}}$$
$$\implies (\mathbf{Y}^{\geq i} \not\perp\!\!\!\perp X_i \mid \mathbf{Z}_i \cup \mathbf{H}_{i-1})_{\mathcal{G}_{\mathrm{pbd}}^{X_i,\mathbf{Y}^{\geq i}}}. \quad (A.7)$$

---

***Proof of Step 1.*** We will prove that any d-connected path between $X_i$ and $Y \in \mathbf{Y}^{\geq i}$ given $\mathbf{Z}_i \cup \mathbf{H}_{i-1}$ in $(\mathcal{G}_{\overline{\mathbf{X}^{\geq i+1}}})_{\mathrm{pbd}}^{X_i,\mathbf{Y}^{\geq i}}$ is preserved in $\mathcal{G}_{\mathrm{pbd}}^{X_i,\mathbf{Y}^{\geq i}}$. First, the d-connected path between $X_i$ and $Y$ in $(\mathcal{G}_{\overline{\mathbf{X}^{\geq i+1}}})_{\mathrm{pbd}}^{X_i,\mathbf{Y}^{\geq i}}$ must be a non-directed path in $\mathcal{G}$, since the path cannot contain incoming edges to any vertices in $\mathbf{X}^{\geq i+1}$; otherwise, it would be cut. As the operation applied to $\mathcal{G}_{\mathrm{pbd}}^{X_i,\mathbf{Y}^{\geq i}}$ does not cut such path, the d-connection of the path remains intact in $\mathcal{G}_{\mathrm{pbd}}^{X_i,\mathbf{Y}^{\geq i}}$, which completes the proof. □

Second, we will witness the following:

---

**Step 2.** For any $\mathbf{Z}_i \subseteq \mathbf{Z} \setminus \mathrm{De}_{\mathcal{G}}(\mathbf{X}^{\geq i+1})$,

$$\left(\mathbf{Y}^{\geq i} \not\perp\!\!\!\perp X_i \mid \mathbf{Z}_i \cup \mathbf{H}_{i-1}\right)_{\mathcal{G}_{\mathrm{pbd}}^{X_i,\mathbf{Y}^{\geq i}}}$$

$$\implies \left(\mathbf{Y}^{\geq i} \not\perp\!\!\!\perp X_i \mid \mathbf{Z}_i \cup \mathbf{H}_{i-1} \cup \mathbf{Z}^{\geq i+1}\right)_{\mathcal{G}_{\mathrm{pbd}}^{X_i,\mathbf{Y}^{\geq i}}}. \tag{A.8}$$

---

***Proof of Step 2.*** The left-hand side of Eq. (A.8) implies that there exists a d-connecting path between $X_i$ and $Y \in \mathbf{Y}^{\geq i}$ given $\mathbf{Z}_i \cup \mathbf{H}_{i-1}$ in $\mathcal{G}_{\mathrm{pbd}}^{X_i,\mathbf{Y}^{\geq i}}$. The path cannot be a directed path, since it is a d-connected path in $\mathcal{G}_{\mathrm{pbd}}^{X_i,\mathbf{Y}^{\geq i}}$, and by the mechanism of $\mathrm{PT}_{\mathcal{G}}^{\mathbf{X}}$. Therefore, the path is either a divergent path $X_i \leftarrow\!\circ\!\rightarrow Y$, or a colliding path $X_i \ast\!\!\rightarrow\!\bullet\!\leftarrow\!\ast Y$ where the collider is an ancestor of some vertices in $\mathbf{Z}_i \cup \mathbf{H}_{i-1}$ (indeed, the collider is an ancestor of $\mathbf{Z}_i$ only, since it cannot be an ancestor of $\mathbf{H}_{i-1}$ because it would create a directed path from $X_i$ to any vertex in $\mathbf{H}_{i-1}$, which contradicts our setting). For the sake of contradiction, suppose there exists a partition of $\mathbf{Z}$ and an topological order of $\mathbf{X}$ such that $\left(\mathbf{Y}^{\geq i} \not\perp\!\!\!\perp X_i \mid \mathbf{Z}_i \cup \mathbf{H}_{i-1}\right)_{\mathcal{G}_{\mathrm{pbd}}^{X_i,\mathbf{Y}^{\geq i}}}$ holds, but

$$\left(\mathbf{Y}^{\geq i} \perp\!\!\!\perp X_i \mid \mathbf{Z}_i \cup \mathbf{H}_{i-1} \cup \mathbf{Z}^{\geq i+1}\right)_{\mathcal{G}_{\mathrm{pbd}}^{X_i,\mathbf{Y}^{\geq i}}}. \tag{A.9}$$

That is, every d-connecting path between $X_i$ and $Y$ given $\mathbf{Z}_i \cup \mathbf{H}_{i-1}$ in $\mathcal{G}_{\mathrm{pbd}}^{X_i,\mathbf{Y}^{\geq i}}$ is blocked by additionally conditioning on $\mathbf{Z}^{\geq i+1}$.

Suppose the above path is a divergent path $X_i \leftarrow\!\circ\!\rightarrow Y$. Then we can consider two cases; the path contains subpath $Z_2 \multimap\!\rightarrow X_i$ or $Z_2 \multimap\!\rightarrow Y$ where $Z_2 \in \mathbf{Z}^{\geq i+1}$. Since we have a full degree-of-freedom of choosing $\mathbf{Z} := (\mathbf{Z}_i : i = 1, \cdots, m)$ where each $\mathbf{Z}_i$ is non-descendant of $\mathbf{X}^{\geq i+1}$ in $\mathcal{G}$, only reason that the vertex $Z_2$ would be in $\mathbf{Z}_k \in \mathbf{Z}^{\geq i+1}$ and not in $\mathbf{Z}_i$ is that it is a descendant of some $X_p \in \mathbf{X}^{\geq i+1}$. Therefore, $Z_2 \multimap\!\rightarrow X_i$ means that there is a directed path $X_p \multimap\!\rightarrow X_i$. However, it contradicts the valid topological order of $\mathbf{X}$, since $X_p \in \mathbf{X}^{\geq i+1}$. On the one hand, suppose the above path be a divergent path $X_i \leftarrow\!\circ\!\rightarrow Y$ that contains a subpath $Z_2 \multimap\!\rightarrow Y$. This means that there exists a directed path from $X_p$ to $Y \in \mathbf{Y}^{\geq p}$. This case also contradicts SAC condition in Eq. (11) because $Z_2$ lies on the proper causal path. Therefore, the path cannot be a divergent path.

Suppose the path is a colliding path $X_i \ast\!\!\rightarrow\!\bullet\!\leftarrow\!\ast Y$ where the collider is an ancestor of some vertices in $\mathbf{H}_{i-1}$. In order for the path to be blocked by conditioning on $\mathbf{Z}^{\geq i+1}$, there must be $Z_2 \in \mathbf{Z}^{\geq i+1}$ on the path that falls into one of the following cases:

  Case 1. There exists a subpath $Z_2 \multimap\!\rightarrow X_i$ in $\mathcal{G}_{\mathrm{pbd}}^{X_i,\mathbf{Y}^{\geq i}}$.

  Case 2. The vertex $Z_2$ is an ancestor of $\mathbf{H}_{i-1}$ in $\mathcal{G}_{\mathrm{pbd}}^{X_i,\mathbf{Y}^{\geq i}}$.

  Case 3. The vertex $Z_2$ is an ancestor of $Y$ in $\mathcal{G}_{\mathrm{pbd}}^{X_i,\mathbf{Y}^{\geq i}}$.

The existence of $Z_2$ even with the full freedom of choosing any partition of $\mathbf{Z}$ means that there exists a directed path from $X_p$ to $Z_2$ as discussed. Therefore, Case 1 implies existence of any directed path from $X_p$ to $X_i$, which contradicts the valid topological order of $\mathbf{X}$.

Case 2 is impossible because it implies the existence of a directed path from $Z_2$ to any vertex in $\mathbf{H}_{i-1}$ (again, a directed path from $X_p$ to the vertex in $\mathbf{H}_{i-1}$), which would results in an invalid topological order.

Finally, Case 3 is also impossible as it implies a directed path from $X_p$ to $Y$, indicating that $Z_2$ lies on the proper causal path.

Therefore, every case leads to a contradiction, demonstrating Eq. (A.8) holds.    □

Combining Step 1 and Step 2, what we have shown is the following:

Step 1 + Step 2: For any $\mathbf{Z}_i \subseteq \mathbf{Z} \setminus \mathrm{De}_{\mathcal{G}}(\mathbf{X}^{\geq i+1})$,

$$(\mathbf{Y}^{\geq i} \not\perp\!\!\!\perp X_i \mid \mathbf{Z}_i \cup \mathbf{H}_{i-1})_{(\mathcal{G}_{\overline{\mathbf{X}^{\geq i+1}}})^{X_i, \mathbf{Y}^{\geq i}}_{\mathrm{pbd}}}$$

$$\implies (\mathbf{Y}^{\geq i} \not\perp\!\!\!\perp X_i \mid \mathbf{Z}_i \cup \mathbf{H}_{i-1} \cup \mathbf{Z}^{\geq i+1})_{\mathcal{G}^{X_i, \mathbf{Y}^{\geq i}}_{\mathrm{pbd}}} \qquad (A.10)$$

We will now show that Eq. (A.6) holds. Let the left-hand side of Eq. (A.6) holds; i.e., there exists a d-connected path between $X_i$ and $Y \in \mathbf{Y}^{\geq i}$ given $\mathbf{Z}_i \cup \mathbf{H}_{i-1}$ in $(\mathcal{G}_{\overline{\mathbf{X}^{\geq i+1}}})^{X_i, \mathbf{Y}^{\geq i}}_{\mathrm{pbd}}$. We have established that such d-connected path is preserved in $\mathcal{G}^{X_i, \mathbf{Y}^{\geq i}}_{\mathrm{pbd}}$ conditioning on $\mathbf{Z}_i \cup \mathbf{H}_{i-1} \cup \mathbf{Z}^{\geq i+1}$, as shown by Eq. (A.10). If this path does not contain any vertices in $\mathbf{X} \setminus \{X_i\}$, then the path is not affected by the operation applied to $\mathcal{G}^{\mathbf{X}, \mathbf{Y}}_{\mathrm{pbd}}$, which completes the proof. Otherwise, even though the path contains vertices in $\mathbf{X} \setminus \{X_i\}$, if the subpath between $X_a \in \mathbf{X} \setminus \{X_i\}$ and any vertex in $\mathbf{Y}^{\geq i}$ is not a proper causal path to $\mathbf{Y}^{\geq i}$, then the path is not affected by the operation applied to $\mathcal{G}^{\mathbf{X}, \mathbf{Y}}_{\mathrm{pbd}}$, which completes the proof.

Suppose the path contains a subpath that is a proper causal path from $X_a \in \mathbf{X} \setminus \{X_i\}$ to $Y$. Then such $X_a$ should be included in $\mathbf{X}^{\geq i+1}$ by the mechanism of $\mathrm{PT}^{\mathbf{X}}_{\mathcal{G}}$. Then, as the path between $X_a$ and $X_i$ cannot be directed in either side, the path between $X_a$ and $X_i$ is either a divergent path $X_a \leftarrow\circ\rightarrow X_i$ or a colliding path $X_a \ast\!\!\rightarrow\bullet\leftarrow\!\!\ast X_i$. Furthermore, since there is no incoming edge to $X_a$, the path should be $X_a \rightarrow\bullet\leftarrow\!\!\ast X_i$. Additionally, the path is d-connected when conditioning on $\mathbf{Z}_i \cup \mathbf{H}_{i-1}$. This implies that there exists a directed path from $X_a \in \mathbf{X}^{\geq i+1}$ to any vertex in $\mathbf{Z}_i \cup \mathbf{H}_{i-1}$, which contradicts our partitioning mechanism $\mathrm{PT}^{\mathbf{X}}_{\mathcal{G}}$. Therefore, there must be no proper causal path from $X_a$ to $Y$.

By proving that Eq. A.6 holds, we have completed the proof that the AC implies the SAC.

## A.4  Proof of Theorem 3

**Lemma A.1** ((van der Zander et al., 2014, Lemma 3.4)). *Let* $\mathbf{X}, \mathbf{Y}, \mathbf{I}, \mathbf{R}$ *be sets of vertices with* $\mathbf{I} \subseteq \mathbf{R}$, $\mathbf{R} \cap (\mathbf{X} \cup \mathbf{Y}) = \emptyset$. *If there exists a d-separator* $\mathbf{Z}_0$ *with* $\mathbf{I} \subseteq \mathbf{Z}_0 \subseteq \mathbf{R}$ *in* $\mathcal{G}$ *then* $\mathbf{Z} = \mathrm{An}_{\mathcal{G}}(\mathbf{X} \cup \mathbf{Y} \cup \mathbf{I}) \cap \mathbf{R}$ *is a d-separator.*

**Lemma A.2** (Construction). *Let* $(\mathbf{X}, \mathbf{Y})$ *denote a disjoint pair of ordered sets. For any* $i \in \{1, 2, \cdots, m\}$, *suppose each* $\mathbf{Z}_j \in (\mathbf{Z}_1, \cdots, \mathbf{Z}_{i-1})$ *is non-descendant of* $\mathbf{X}^{\geq j+1}$ *and disjoint to* $(\mathbf{X}, \mathbf{Y})$. *Let* $\mathbf{Z}^{\mathrm{an}}_i$ *be an ordered set in Eq.* (17) *defined with respect to* $\mathbf{Z}^{(i-1)}$, *which is denoted as* $\mathbf{Z}^{\mathrm{an}}_i(\mathbf{Z}^{(i-1)})$. *Then, the following statements are equivalent.*

1. $\mathbf{Z}^{\mathrm{an}}_i(\mathbf{Z}^{(i-1)})$ *is non-descendant of* $\mathbf{X}^{\geq i+1}$ *satisfying Eqs.* (10, 11).

2. *There exists a set* $\mathbf{Z}_i \subseteq \mathbf{V} \setminus (\mathbf{X} \cup \mathbf{Y} \cup \mathbf{H}_{i-1} \cup \mathrm{De}_{\mathcal{G}}(\mathbf{X}^{\geq i+1}))$ *satisfying Eqs.* (10, 11).

3. *There exists a set* $\mathbf{Z}_i \subseteq \mathbf{V} \setminus (\mathbf{X} \cup \mathbf{Y} \cup \mathbf{H}_{i-1} \cup \mathrm{De}_{\mathcal{G}}(\mathbf{X}^{\geq i+1}))$ *satisfying the following:*

$$P(\mathbf{y}^{\geq i} \mid \mathrm{do}(\mathbf{x}^{\geq i}), \mathbf{h}_{i-1}) = \sum_{\mathbf{z}_i} P(\mathbf{y}^{\geq i+1} \mid \mathrm{do}(\mathbf{x}^{\geq i+1}), \mathbf{h}_i) P(\mathbf{z}_i \mid \mathbf{h}_{i-1}) P(\mathbf{y}_i \mid x_i, \mathbf{z}_i, \mathbf{h}_{i-1}).$$

***Proof of Lemma A.2.*** By Theorem 1, $(2) \Leftrightarrow (3)$ holds. Additionally, $(1) \implies (3)$ by Lemma S.1, which is equivalent to $(1) \implies (2)$. Therefore, we only need to prove $(2) \implies (1)$.

We apply Lemma A.1 by replacing $\mathcal{G}$ in Lemma A.1 as $\mathcal{G}^{\mathbf{X}, \mathbf{Y}}_{\mathrm{psbd}, i}$, $\mathbf{X}$ in Lemma as $\{X_i\}$, $\mathbf{Y}$ in Lemma as $\mathbf{Y}^{\geq i}$, $\mathbf{I}$ in Lemma as $\mathbf{H}_{i-1}$, $\mathbf{Z}_0$ in Lemma as an union of $\mathbf{H}_{i-1}$ and a certain $\mathbf{Z}_i$ satisfying Eqs. (10, 11), and $\mathbf{R}$ as an union of $\mathbf{H}_{i-1}$ and $\mathbf{V} \setminus (\mathbf{X} \cup \mathbf{Y} \cup \mathbf{H}_{i-1} \cup \mathrm{dpcp}_{\mathcal{G}}(X_i, \mathbf{Y}^{\geq i}) \cup \mathrm{De}_{\mathcal{G}}(\mathbf{X}^{\geq i+1}))$. Then, Lemma A.1 states that, under the statement $(2)$, there exists a d-separator between $X_i$ and $\mathbf{Y}^{\geq i}$, which is $\mathbf{Z}^{\mathrm{an}}_i \cup \mathbf{H}_{i-1} := \left( \mathrm{An}_{\mathcal{G}^{\mathbf{X}, \mathbf{Y}}_{\mathrm{psbd}, i}}(\{X_i\} \cup \mathbf{Y}^{\geq i} \cup \mathbf{H}_{i-1}) \setminus \mathbf{F}_i \right) \cup \mathbf{H}_{i-1}$. This completes the proof. $\square$

We emphasize that Lemma A.2 is stated for a fixed $\mathbf{Z}^{(i-1)}$; i.e., $\mathbf{Z}^{\mathrm{an}}_i$ in Lemma A.2 is dependent on some specific choice of $\mathbf{Z}^{(i-1)}$. This dependency is highlighted by the notation $\mathbf{Z}^{\mathrm{an}}_i(\mathbf{Z}^{(i-1)})$.

Then, the remaining part of the proof is to show that, for any given $\mathbf{Z}^{(i-1)}$, if $\mathbf{Z}_i$ satisfies Eqs. (10, 11), then $\mathbf{Z}_i^{\mathrm{an}}(\mathbf{Z}_1^{\mathrm{an}}, \cdots, \mathbf{Z}_{i-1}^{\mathrm{an}})$ satisfies Eqs. (10, 11). To show through contradiction, suppose $\mathbf{Z}_i^{\mathrm{an}}(\mathbf{Z}_1^{\mathrm{an}}, \cdots, \mathbf{Z}_{i-1}^{\mathrm{an}})$ does not satisfy Eqs. (10, 11), even if $\mathbf{Z}_i$ satisfies them given $\mathbf{Z}^{(i-1)}$. By Lemma A.2, this implies that, given $\{\mathbf{Z}_1^{\mathrm{an}}, \cdots, \mathbf{Z}_{i-1}^{\mathrm{an}}\}$, no $\mathbf{Z}_i' \subseteq \mathbf{V} \setminus \mathbf{F}_i^{\mathrm{an}}$ satisfies Eqs. (10, 11).

However, consider the following choice of $\mathbf{Z}_i'$:

$$\mathbf{Z}_i' := (\mathbf{Z}_i \cup \mathbf{Z}^{(i-1)}) \setminus \{\mathbf{Z}_1^{\mathrm{an}}, \cdots, \mathbf{Z}_{i-1}^{\mathrm{an}}\}.$$

Then, $\mathbf{Z}_i' \cap \mathrm{dpcp}_{\mathcal{G}}(X_i, \mathbf{Y}^{\geq i}) = \emptyset$, since $\mathbf{Z}_i \cap \mathrm{dpcp}_{\mathcal{G}}(X_i, \mathbf{Y}^{\geq i}) = \emptyset$ and every set in $\mathbf{Z}^{(i-1)}$ is non-descendant of $X_i$. Also,

$$(X_i \perp\!\!\!\perp \mathbf{Y}^{\geq i} \mid \mathbf{Z}_i' \cup \mathbf{H}_i^{\mathrm{an}})_{\mathcal{G}_{\mathrm{psbd},i}^{\mathbf{X},\mathbf{Y}}} = (X_i \perp\!\!\!\perp \mathbf{Y}^{\geq i} \mid \mathbf{Z}_i \cup \mathbf{H}_i)_{\mathcal{G}_{\mathrm{psbd},i}^{\mathbf{X},\mathbf{Y}}}$$

where the right-hand side independence $(X_i \perp\!\!\!\perp \mathbf{Y}^{\geq i} \mid \mathbf{Z}_i \cup \mathbf{H}_i)_{\mathcal{G}_{\mathrm{psbd},i}^{\mathbf{X},\mathbf{Y}}}$ holds by the given assumption. This contradicts that no $\mathbf{Z}_i' \subseteq \mathbf{V} \setminus \mathbf{F}_i^{\mathrm{an}}$ satisfies Eqs. (10, 11). The contradiction happens since we assumed that $\mathbf{Z}_i^{\mathrm{an}}(\mathbf{Z}_1^{\mathrm{an}}, \cdots, \mathbf{Z}_{i-1}^{\mathrm{an}})$ does not satisfy Eqs. (10, 11).

Therefore, we conclude that, for any $i \in \{1, 2, \cdots, m\}$, $\mathbf{Z}_i^{\mathrm{an}}(\mathbf{Z}_1^{\mathrm{an}}, \cdots, \mathbf{Z}_{i-1}^{\mathrm{an}})$ satisfies Eqs. (10, 11) whenever $\mathbf{Z}_i(\mathbf{Z}^{(i-1)})$ satisfies Eqs. (10, 11). This completes the proof.

### A.5    Proof of Theorem 4

For each $i \in \{1, 2, \cdots, m\}$, $\mathbf{Z}_i^{\min}$ is the minimal d-separator relative to $(X_i, \mathbf{Y}^{\geq i})$ in $\mathcal{G}_{\mathrm{psbd},i}^{\mathbf{X},\mathbf{Y}}$ by (van der Zander and Liśkiewicz, 2020, Proposition 5.2); i.e., excluding any vertices from $\mathbf{Z}_i^{\min}$ violates the d-separability, and $\mathbf{Z}_i^{\min}$ is a valid d-separator whenever $\mathbf{Z}_i^{\mathrm{an}}$ is a valid d-separator relative to $(X_i, \mathbf{Y}^{\geq i})$ in $\mathcal{G}_{\mathrm{psbd},i}^{\mathbf{X},\mathbf{Y}}$[4].

---

[4]The code is available at `https://github.com/snu-causality-lab/minSAC`

