# OpenReview forum: "Complete Graphical Criterion for Sequential Covariate Adjustment in Causal Inference"
_NeurIPS.cc/2024/Conference — NeurIPS 2024 poster_

### Official Review · Reviewer_aPdz · 2024-07-03

**Soundness:** 3
**Presentation:** 2
**Contribution:** 3
**Rating:** 6
**Confidence:** 4

**Summary:**

The paper addresses the incompleteness of current sequential covariate adjustment criteria in causal inference. It introduces a sound and complete graphical criterion for sequential covariate adjustment, termed Sequential Adjustment Criterion (SAC), and provides an algorithm for identifying a minimal sequential covariate adjustment set. This work demonstrates the limitations of existing criteria, proposes the SAC, and develops a method to construct and optimize adjustment sets for efficient causal effect estimation.

**Strengths:**

The paper demonstrates the limitations of the multi-outcome Sequential Back-Door (mSBD) criterion by presenting examples where mSBD fails to identify causal effects that can be identified using sequential covariate adjustment.

The paper presents a new criterion, SAC, which is both sound and complete for sequential covariate adjustment.

Develops an algorithm to find the minimal sequential covariate adjustment set, reducing unnecessary computations.

**Weaknesses:**

The presentation of the manuscript is narrow and difficult to follow. For example, in Eq. 2, the subscript should be used to distinguish the orders of variables, specially, in the left-hand side of the equation.

Definition 3 seems incorrect since the presentation 'Z is said to be a sequential adjustment set relative to (X,Y) in G if…' is difficult to understand. How to determine a sequential adjustment set from G?

How to understand ``the causal effect is given as an adjustment''?

The conclusion in Definition 4 is not readable, please double-check.

What type of causal graph satisfies the proposition 3? Can you provide a type to summarize this kind of causal graph?

The paper primarily provides theoretical examples and proofs to demonstrate the effectiveness of SAC. Further, the causal graph should be given and the lack of extensive validation with real-world data sets might be seen as a limitation.Y

**Questions:**

See Weaknesses

**Limitations:**

Yes

---

> ### Author Rebuttal · Authors · 2024-08-06
>
> Thank you for the valuable feedback.
>
> ---
>
> > For example, in Eq. 2, the subscript should be used to distinguish the orders of variables, specially, in the left-hand side of the equation.
>
> In Eq. (2) included in Definition 3, $\mathbf{X}$ and $\mathbf{Y}$ are already defined as $(\mathbf{X}_1,\cdots ,\mathbf{X}_m)$ and $(\mathbf{Y}_0,\cdots ,\mathbf{Y}_m)$, respectively.
>
> ---
>
> > __(1)__ Definition 3 seems incorrect since the presentation 'Z is said to be a sequential adjustment set relative to (X,Y) in G if…' is difficult to understand. __(2)__ How to determine a sequential adjustment set from G?
>
>
> 1. That is a standard and established way to express it. Similar expressions can be found in existing works [1, 2].
> 2. Section 4.1 and Section 4.2 address methods for constructing $\mathbf{Z}$, focusing on how to determine the sequential adjustment set.
>
> ---
>
> > How to understand “the causal effect is given as an adjustment”?
>
> It means that the causal effect (left-hand side of Eq. (13)) is expressed as covariate adjustment (right-hand side of Eq. (13)). For further explanation of the adjustment criterion, please refer to lines 73-75 of our paper or the existing works referenced in lines 17-27.
>
> ---
>
> > The conclusion in Definition 4 is not readable
>
> It would be greatly appreciated if you could specify which part you find "not readable," so that we can better address your comment.
>
> ---
>
> > What type of causal graph satisfies the proposition 3? Can you provide a type to summarize this kind of causal graph?
> >
>
> In lines 142-154, we illustrate such an example immediately following Proposition 3, where mSBD is not satisfied, but the causal effect can still be expressed by SCA.
>
> ---
>
> > Further, the causal graph should be given and the lack of extensive validation with real-world data sets might be seen as a limitation.Y
>
> Our paper falls into the causal effect identification category in which the key task is to express the causal effect as a function of observational distribution using assumptions encoded in a causal graph. Therefore, we disagree that the presence of causal graphs is a limitation of our work.
>
> As you noted, our work is theory-oriented, presenting a complete criterion for sequential covariate adjustment and introducing an algorithm for constructing a (minimal) sequential adjustment set. Our focus has been on clearly elucidating the theories and algorithms through appropriate examples. We believe that simulation studies or experimental validations would not enhance our theoretical work. However, to demonstrate the practical benefits of our theories, we will provide implementation code during the revision process.
>
>
> ---
>
> <Reference>
>
> [1] Pearl, J. (1995). Causal diagrams for empirical research. Biometrika, 82(4):669–710.
>
> [2] Pearl, J. (2000). Causality: Models, Reasoning, and Inference. Cambridge University Press, New York. 2nd edition, 2009.

---

> > ### Comment · Reviewer_aPdz · 2024-08-12
> > **Thank you for the rebuttal.**
> >
> > Thank you for the rebuttal. My concerns have been partially addressed.
> >
> > It would be better to add some examples for illustrating Definition 4.
> >
> > I am not confident about whether we can summary the graph that mSBD criterion fail.

---

> > > ### Author Response · Authors · 2024-08-13
> > >
> > > > It would be better to add some examples for illustrating Definition 4.
> > >
> > > Please note that Definition 4 (Partitioning Operator) has already been applied to all subsequent examples.
> > >
> > > ---
> > >
> > > > I am not confident about whether we can summary the graph that mSBD criterion fail.
> > >
> > > What do you mean by _summary the graph that mSBD criterion fail_ ?  In this response, we presume that the reviewer is concerned with the case where the mSBD criterion fails while the proposed SAC succeeds (i.e., a type of causal graph satisfying Proposition 3).
> > > First, we recall that SAC implies mSBD criterion as follows:
> > > $$
> > > (\mathbf{Y}^{\geq i+1} \perp X_i \mid  \mathbf{Z}_i, \mathbf{H}\_{i-1} ){\mathcal{G}^{\mathbf{X},\mathbf{Y}}\_{\operatorname{psbd},i}}
> > > \implies
> > > (\mathbf{Y}^{\geq i+1} \perp X_i \mid  \mathbf{Z}_i, \mathbf{H}\_{i-1} )\_{{\underline{X_i}}\overline{{\mathbf{X}^{\geq i+1}}}},
> > > $$
> > > since (1) $\mathcal{G}^{\mathbf{X},\mathbf{Y}}\_{\operatorname{psbd},i}$ cuts fewer edges than $\mathcal{G}\_{{\underline{X_i}}\overline{{\mathbf{X}^{\geq i+1}}}}$, and (2) removing more edges from $\mathcal{G}^{\mathbf{X},\mathbf{Y}}\_{\operatorname{psbd},i}$ to obtain $\mathcal{G}\_{{\underline{X_i}}\overline{{\mathbf{X}^{\geq i+1}}}}$ doesn’t decrease any d-separations. This means that it’s impossible for $\mathbf{Z}$ to satisfy SAC while violating mSBD. Equivalently, Whenever mSBD holds, SAC will also hold simultaneously.
> > > This observation allows us to come up with a scenario in which SAC holds but mSBD fails when there exists $\mathbf{Z}_i$ that is a descendant of $\mathbf{X}_i$. We illustrated this scenario in lines 142-154. Specifically, in Fig. 2a, $\mathbf{Z}_2 := \\{Z_c,Z_d\\}$ does not satisfy the mSBD criterion since it’s a descendant of $X_2$, while it does satisfy SAC.

---

> > > > ### Comment · Reviewer_aPdz · 2024-08-13
> > > >
> > > > Thank you for your further clarification. I am happy to increase my rating from 5 to 6.

---

> > > > > ### Author Response · Authors · 2024-08-13
> > > > >
> > > > > We sincerely appreciate your encouragement of thoughtful discussion on points that other reviewers might also have been concerned about!

---

### Official Review · Reviewer_VqEB · 2024-07-08

**Soundness:** 3
**Presentation:** 2
**Contribution:** 3
**Rating:** 6
**Confidence:** 4

**Summary:**

The paper investigates the problem of identifying total causal effects via sequential covariate adjustments, which generalizes the standard, well-studied covariate adjustment. Unlike the standard static case, where there exists sound and complete graphical identification criterion, for the sequential counterpart only sound but incomplete criterion is known. The incompleteness is demonstrated in the submitted work. The main achievement of the paper is the sequential (graphical) adjustment criterion which is sound and complete for sequential covariate adjustment that is much more involved, but more powerful than in the standard static adjustment.

**Strengths:**

The research of this work is well motivated and concerns an important problem in causality. The task to identify the total causal effects via covariate adjustments in well studied and the authors improve the previous results in this area presenting the first sound and complete graphical criterion for the sequential covariate adjustment which nicely extends the sound and complete criterion for the standard case.

To prove the main result, the authors extend in an elegant way the constructive adjustment criterion by (van der Zander et al., 2014) based on the adjustment criterion proposed in (Shpitser et al., 2010). A nice technical result is a construction of sequential adjustment set (in Theorem 3).

**Weaknesses:**

The presentation of the paper needs improvement. A number of definitions and formulas require clarification and improving of mathematical precision. For example, in Definition 3 the authors consider H_i and in Eq. (2) they use in the formula h_{j-1}. So, what is the definition of h_{-1} and of h_{0}? Note that according to the definition of H_i, for H_{-1}, the sets X^{(-1)} and Y^{(-1)} are undefined, and for H_{0}, the set X^{(0)} is undefined. It seems that the authors mean, in such cases H_i are empty. Also, some sets Y^{(i)} can be empty. These issues should be clarified.

Moreover, it is not clear what is the relationship between the sequential covariate adjustment formula presented in Definition 3 and the corresponding formula in (Jung et al., 2020, Theorem 1) used to identify causal effect by mSBD adjustment. In the submitted paper the authors present in Proposition 2 (mSBD adjustment (Jung et al., 2020)) formula (2) that differs from the original mSBD adjustment formula given in (Jung et al., 2020) in Theorem 1 as Eq. (4). So, to what extent is the incompleteness of mSBD criterion presented in Proposition 3 justified?  For more remarks, see paragraph "question" below for details.

The authors do not provide experimental results verifying the performance of the proposed algorithms.

**Questions:**

In Eq. (2) x_j and z_j are undefined for j=0.

In Eq. (2) z_{j+1} is undefined for j=m.

Write explicitly how do you define sets H_i in Definition 5. In the same way as n Definition 3?

Why do you consider in Definition 3, 4, etc. in the sequence X = (X_1, . . . ,X_m) the elements X_i as sets? In fact every X_i denotes here a single variable.

Eq. (7) is not correct, resp., it does not correspond to the definition given in Eq. (2) in Definition 3 as well as formula used in Eq. (6). Specifically, in the factor P(y2 | x1, x2, z1, z2) in Eq. (7) variable y1 in the conditioning set is missing. Note that in this example we have: Y1 = {Y1}.

The formula presented in Eq. (2) for sequential covariate adjustment is different than the formula in (Jung et al., 2020, Theorem 1) used to identify causal effect by mSBD adjustment. Could you show that they are equivalent?

The next issue concerns Theorem 3. The authors define there in Eq. (15) the forbidden sets F_i using in the right-hand  side the sets H_{i-1}. Again, the sets H_{i} should be defined. E.g. if H_i := X^{(i)} ∪ Y^{(i)} ∪ Z^{(i)} -- as assumed in Definition 3 --  then X ∪ Y ∪ H_{i−1} in (15) is equal to X ∪ Y ∪ Z^{(i)}. So why do you use  H_{i−1} in (15)?

In Eq. (16) you use union of sets Z_{j}^{an} for j=1,..., i-1. For i=1 this summation makes no sense.

In Definition 7: (Sequential Adjustment Criterion) --> (Sequential Adjustment Criterion (SAC))

**Limitations:**

yes

---

> ### Author Rebuttal · Authors · 2024-08-06
>
> Thank you for the valuable feedback and detailed comments.
>
> ---
>
> > For example, in Definition 3 the authors consider H_i and in Eq. (2) they use in the formula h_{j-1}. So, what is the definition of h_{-1} and of h_{0}? Note that according to the definition of H_i, for H_{-1}, the sets X^{(-1)} and Y^{(-1)} are undefined, and for H_{0}, the set X^{(0)} is undefined. It seems that the authors mean, in such cases H_i are empty. Also, some sets Y^{(i)} can be empty. These issues should be clarified.
>
> Thank you for your feedback. We implicitly defined a set whose indices are outside their original range as an empty set. For example,
>
> 1. In Eq. (2), all the subscripted values whose indices are out of range (e.g., $\mathbf{x}\_0$, $\mathbf{z}\_0$ and $\mathbf{z}\_{m+1}$ are considered as an empty set.
>
>
> 2. In Eq. (16), it implies that we go over index j such that $1 \leq j \leq i-1$. Hence, if $i=1$ implies that there is no union, i.e., an empty set will be returned for the term.
>
> We will explicitly state this in the revised version of the paper.
>
> ---
>
> > The formula presented in Eq. (2) for sequential covariate adjustment is different than the formula in (Jung et al., 2020, Theorem 1) used to identify causal effect by mSBD adjustment. Could you show that they are equivalent?
>
> Here is the proof of the equivalence, as derived in line 393 in the Appendix:
>
> $\underbrace{P_\mathbf{x}(\mathbf{y}) = \sum_\mathbf{z} \prod\_{j=0}^n P(\mathbf{y}_j \mid \mathbf{x}^{(j)},\mathbf{z}^{(j)},\mathbf{y}^{(j-1)})\prod\_{k=1}^n P(\mathbf{z}_k \mid \mathbf{x}^{(k-1)},\mathbf{z}^{(k-1)},\mathbf{y}^{(k-1)})}\_{ \text{Jung et al.,2020, Thm1}}$
>
> $\qquad \quad = \sum\_\mathbf{z} \prod\_{j=0}^n P(\mathbf{y}_j \mid \mathbf{x}_j,\mathbf{z}_j,\mathbf{h}\_{j-1})\prod\_{k=1}^n P(\mathbf{z}_k \mid \mathbf{h}\_{k-1})$
>
> $\qquad \quad =\sum_\mathbf{z} \prod\_{j=0}^n P(\mathbf{y}_j \mid \mathbf{x}_j,\mathbf{z}_j,\mathbf{h}\_{j-1})P(\mathbf{z}\_{j+1} \mid \mathbf{h}_j)$
>
> $\qquad \quad =\underbrace{\sum\_\mathbf{z} \prod\_{j=0}^n P(\mathbf{z}\_{j+1},\mathbf{y}_j \mid \mathbf{h}\_{j-1},\mathbf{x}_j,\mathbf{z}_j,) = P(\mathbf{y} \mid do(\mathbf{x}))}\_{\text{Ours Eq. (2)}}$
>
> ---
>
> > Write explicitly how do you define sets H_i in Definition 5. In the same way as n Definition 3?
>
> Yes. We will invoke the definition of $\mathbf{H}_i$ in Def. 3 to Def. 5.
>
>
>
> ---
>
> > Why do you consider in Definition 3, 4, etc. in the sequence X = (X_1, . . . ,X_m) the elements X_i as sets? In fact every X_i denotes here a single variable.
>
> This is because a set serves as a basic unit for graphical operators like \(Pa\), \(An\), and so on. Furthermore, by treating \(X_i\) as a set, we can maintain consistency in handling \(X_i\), \(\mathbf{Y}_i\), and \(\mathbf{Z}_i\) simultaneously.
>
> ---
>
> > Eq. (7) is not correct, resp., it does not correspond to the definition given in Eq. (2) in Definition 3 as well as formula used in Eq. (6). Specifically, in the factor P(y2 | x1, x2, z1, z2) in Eq. (7) variable y1 in the conditioning set is missing. Note that in this example we have: Y1 = {Y1}.
>
> Thank you for catching these typos. We will make the necessary corrections.
>
> ---
>
> >  then X ∪ Y ∪ H_{i−1} in (15) is equal to X ∪ Y ∪ Z^{(i)}. So why do you use H_{i−1} in (15)?
>
> We use $\mathbf{H}\_{i-1}$ in (15) to main comprehensibility. Specifically, Equation (15) is read as follow -- the forbidden set is an union of (1) treatments, (2) outcomes, (3) predecessors ($\mathbf{H}\_{i-1}$), (4) dpcp, and (5) a descendent set.
>
>
> ---
>
> > In Definition 7: (Sequential Adjustment Criterion) --> (Sequential Adjustment Criterion (SAC))
>
> We will revise as suggested. Thanks.

---

> > ### Comment · Reviewer_VqEB · 2024-08-13
> > **Comment**
> >
> > Thanks to the authors for all the answers that address my concerns.

---

> > > ### Author Response · Authors · 2024-08-13
> > >
> > > We are pleased that we have addressed all of your concerns. Thank you for taking the time to offer your valuable and detailed feedback.

---

### Official Review · Reviewer_sfA3 · 2024-07-10

**Soundness:** 3
**Presentation:** 3
**Contribution:** 2
**Rating:** 6
**Confidence:** 3

**Summary:**

To estimate causal effects given observational data, previous works provide graphical criterion that is not complete in the sequential and multi-outcome cases. The following work extends the complete adjustment criterion to these cases. The non-completeness of previous sequential and multi-outcome criterion is highlighted and then a complete graphical criterion is proposed. The authors then propose an algorithm that finds the minimum adjustment set that satisfies SAC relative to some treatment and outcome in the graph.

**Strengths:**

- The paper is clearly written, the problem being solved is clear as well as the limitations of previous work. The examples also help a lot in this.
- Despite the low novelty (in weakness) this has not been done before and having access to a complete graphical criterion for sequential adjustment is of practical importance.

**Weaknesses:**

- The work does seem to be a simple extension of AC to the sequential case. I'm not sure what the main new insight here is, as the concepts required for SAC seem very simple extensions of AC. The adjustment in AC opens all causal paths and blocks all others, following this logic onto the sequential case seems to imply the criterion given in SAC.

**Questions:**

Questions:
- For an adjustment set Z where mSBD holds, SAC holds as well (corollary 1) but can SAC provide a smaller adjustment set?
- In the example of Figure 3, doesn't SAC collapse to be the same as AC?

Minor comments:
- Definition of proper causal paths is different from previous papers (eg Shipster 2010), would be good to give intuition here. The definition of proper causal path may not be strictly true.
- References for definitions in Section 3?
- H_i should be redefined in Definition 5 to make the definition complete, or refer back to where it is defined

**Limitations:**

This is clear throughout the work.

---

> ### Author Rebuttal · Authors · 2024-08-06
>
> We thank the reviewer for the time and valuable feedback!
>
> ---
>
> > The work does seem to be a simple extension of AC to the sequential case.
>
> Our proposed method is an extension of the AC, similar to how the sequential back-door adjustment extends the back-door adjustment. The principle of our method is to open the proper causal paths and block the proper non-causal paths. However, we disagree that the extension procedure is simple.
>
> As the saying goes “The devil is in the details”, there are nontrivial challenges when extending the AC to the SAC:
>
> 1. Unlike the adjustment case, partitioning the variables is essential. Choosing a proper partition that preserves causal interpretation (e.g., $\mathbf{X}_i$ causes $\mathbf{Y}^{\geq i}$) is nontrivial.
>
> 2. There is a significant gap between conjecturing and actually witnessing the soundness and completeness of SAC. Specifically, the formal approach for witnessing the soundness and completeness of AC proposed by [1] is hardly generalizable to SAC cases. The original proof of AC by [1] uses the counterfactual network [2] graph to show the soundness and completeness of AC. Such a proof technique is hardly generalizable for sequential cases, since, by the nature of the counterfactual network, the number of treatment variables to be considered in the proof increases exponentially with the number of treatments. To circumvent such challenges, we devised a new proof technique that can be commonly applied to both AC and SAC.
>
> 3. One naive approach is to condition the previous set $\mathbf{H}_{i-1}$, and apply the AC with respect to $(X_i, \mathbf{Y}^{\geq i})$ in the graph where the previous set is conditioned on. However, the graph induced by conditioning the previous set may not be a DAG (with bidirected edges), even if the original causal graph is a DAG. This happens when the previous set contains colliders [3].
>
> Circumventing such challenges while adhering to the principle—to open the proper causal paths and block the proper non-causal paths—is a nontrivial problem requiring formal treatments, as we have done in this paper.
>
> ---
>
> > For an adjustment set Z where mSBD holds, SAC holds as well (corollary 1) but can SAC provide a smaller adjustment set?
>
> Just to clarify, please note that mSBD and SAC determine whether a causal effect can be expressed as Sequential Covariate Adjustment (SCA) when $\mathbf{Z}$ is given, rather than constructing $\mathbf{Z}$ itself. Algorithm 1 (minSCA) is designed to provide a smaller (or equal) adjustment set such that no strict subset of it is a valid adjustment set.
>
> For example, consider Figure 4a. Here, $\mathbf{Z} = (\emptyset, \{Z_a, Z_b, Z_c\})$ satisfies the mSBD criterion (and SAC) relative to $((X_1, X_2), (\emptyset, Y))$. However, as discussed in lines 301-308, Algorithm 1 (minSCA) yields $\mathbf{Z}^{min} = (\emptyset, \{Z_a\})$ and this set also satisfies the SAC. This example illustrates that the minSCA procedure can yield a smaller adjustment set.
>
> ---
>
> > In the example of Figure 3, doesn't SAC collapse to be the same as AC?
>
> In the example, the set $\mathbf{Z} = \\{ Z_a,Z_b \\}$ satisfies AC, and $\mathbf{Z} := (\\{Z_a,Z_b\\},\\{\\})$ satisfies SAC. In other words, the set $\mathbf{Z} = \\{ Z_a,Z_b \\}$ is a valid admissible \& sequentially admissible set. However, this observation doesn't hold in general. For example, $\mathbf{Z} := (\\{\\},\\{\\})$ also satisfies the SAC in the same graph. However, the AC is not satisfied, because $\mathbf{Z}$ fails to block the path $Y_2 \to Z_b \to X_2$ in the proper back-door graph $\mathcal{G}_{\text{pbd}}^{\mathbf{X},\mathbf{Y}}$.
>
> ---
>
> > Definition of proper causal paths is different from previous papers (eg Shipster 2010), would be good to give intuition here. The definition of proper causal path may not be strictly true.
>
> Our definition is comparable with the definitions in the previous papers as follow. First, a causal path (equivalently, a _directed path_) is said to be proper if it does not intersect $\mathbf{X}$ except at the end point [1]. Equivalently, the path is _proper_ if only its start node is in $\mathbf{X}$ [4]. These definitions match with our definition in line 78, which states that the path does not contain a node in $\mathbf{X} \setminus \\{ X \\}$. In other words, the path starting from $X$ is not overlapped with $\mathbf{X}$ if and only if the path doesn't contain a node in $\mathbf{X} \setminus \\{ X \\}$. Therefore, our definition is equivalent to the definitions in the previous papers.
>
> ---
>
> > References for definitions in Section 3?
>
> We presume that you are referring to Definition 3 (SCA). We will refer to it just before Definition 3, which we already referenced sequential covariate adjustment in lines 28-29 of the Introduction.
>
> ---
>
> > H_i should be redefined in Definition 5 to make the definition complete, or refer back to where it is defined
>
> Thank you for the suggestion. We first note that $\mathbf{H}_{i-1}$ has been defined in Def. 3. We will invoke this definition in Def. 5.
>
> ---
>
> <Reference>
>
> [1] Shpitser, I., VanderWeele, T., and Robins, J. M. (2010). On the validity of covariate adjustment for estimating causal effects.
>
> [2] Shpitser, I. and Pearl, J. (2008). Complete identification methods for the causal hierarchy. *Journal of Machine Learning Research*, 9:1941–1979.
>
> [3] Richardson, T., & Spirtes, P. (2002). Ancestral graph Markov models. *The Annals of Statistics*, 30(4), 962-1030.
>
> [4] van der Zander, B., Li ́skiewicz, M., and Textor, J. (2014). Constructing separators and adjustment sets in ancestral graphs. In *Proceedings of the UAI 2014 Conference on Causal Inference: Learning and Prediction-Volume 1274*, pages 11–24.

---

> > ### Author Response · Authors · 2024-08-13
> >
> > Thank you again for your detailed comments. The discussion period is nearing its end, with just about a day remaining. Could you please confirm if our rebuttal has adequately addressed your concerns and comments?

---

### Official Review · Reviewer_UrnJ · 2024-07-12

**Soundness:** 3
**Presentation:** 3
**Contribution:** 3
**Rating:** 6
**Confidence:** 2

**Summary:**

This paper develops a complete and constructive criterion for sequential covariate adjustment. An algorithm is provided to identify a minimal sequential covariate adjustment set, which is efficient by ensuring that no unnecessary vertices are included.

**Strengths:**

- The problem considered is important in causal inference.
- The discussion of the limitations of existing criterion is insightful, often illustrated with example.
- The theoretical results and algorithm appear to be coherent and sound, though I am not familiar with the details of covariate adjustment to verify them and go through the proofs.

**Weaknesses:**

- Simulation studies are not provided, though it is fine for a theory-focused paper.
- The computational complexity of the algorithm is not discussed.

**Questions:**

What is the computational complexity of the proposed algorithm?

**Limitations:**

I did not manage to find a clear discussion of the limitations.

---

> ### Author Rebuttal · Authors · 2024-08-06
>
> Thank you for sharing your thoughts and feedback with us.
>
> > Simulation studies are not provided, though it is fine for a theory-focused paper.
>
> As you noted, our work is theory-oriented, presenting a complete criterion for sequential covariate adjustment and introducing an algorithm for constructing a (minimal) sequential adjustment set. Our focus has been on clearly elucidating the theories and algorithms through appropriate examples. We believe that simulation studies or experimental validations would not enhance our theoretical work. However, to demonstrate the practical benefits of our theories, we will provide implementation code during the revision process.
>
> ---
>
> > The computational complexity
>
> 1. The proper sequential back-door graph (in Def. 6) can be constructed in $O(|\mathbf{V}|+|\mathbf{E}|)$ time [1].
>
> 2. Provided the partition of $\mathbf{Z}= (\mathbf{Z}_1,\cdots,\mathbf{Z}_m)$, checking if such $\mathbf{Z}$  satisfies the SAC in Def. 7 takes $O(m|\mathbf{V}| + m|\mathbf{E}|)$, since checking each d-separation takes $O(|\mathbf{V}| + |\mathbf{E}|)$.
>
> 3. Constructing $\mathbf{Z}^{an}_i$ in Theorem 3 takes $O(|\mathbf{V}| + |\mathbf{E}|)$ since finding the set dpcp, descendent and ancestor sets take $O(|\mathbf{V}| + |\mathbf{E}|)$.
>
> 4. As mentioned in line 286, finding closure can be done in $O(|\mathbf{V}| + |\mathbf{E}|)$.
>
> 5. Combining all of these analyses, the computational complexity of the proposed method is $O(m|\mathbf{V}| + m|\mathbf{E}|)$.
>
> ---
>
> <Reference>
>
> [1] van der Zander, B., Li ́skiewicz, M., and Textor, J. (2014). Constructing separators and adjustment sets in ancestral graphs. In Proceedings of the UAI 2014 Conference on Causal Inference: Learning and Prediction-Volume 1274, pages 11–24.

---

> > ### Comment · Reviewer_UrnJ · 2024-08-11
> >
> > Thanks for the responses. I do not have further concerns beyond these, and will retain my score until further discussion with the other reviewers.

---

> > > ### Author Response · Authors · 2024-08-13
> > >
> > > We’re glad to have addressed all the concerns from you and other reviewers, `aPdz` and `VqEB`. Thank you for taking the time to review our work.

---

### Decision · Program_Chairs · 2024-09-25

**Decision:**

Accept (poster)

**Comment:**

The reviewers are highly consistent in their evaluation of the paper’s extension of the adjustment criterion to sequential data: This extension is valid and useful, though not particularly unexpected. The authors note that while the results are not surprising, there are also nontrivial challenges when extending the adjustment criteria to sequential data. Thus, the results are worth publishing.

Several reviewers suggest improvements in the presentation, and the authors should revise their presentation accordingly. Specifically, reviewers VqEB and aPdz note several definitions and formulas that could be made clearer and more precise. In some cases, the authors responses are defensive, but they should take the reviewer’s comments as one indication that a careful reader had questions and misunderstandings and thus some revisions are justified.